# Prognostic stratification in hepatocellular carcinoma using a telomerase-related lncRNA signature derived from TCGA database

Runze Yang[1], Luchao Xing[1], Chenghao Wang[1], Yunhao Zhang[2], Songzhuang Xie[2], Jianlei Yuan ORCID [2]*

**1** Department of Graduate School, Chengde Medical University, Chengde, Hebei, China, **2** Department of Hepatobiliary and Pancreatic Surgery, Cangzhou People's Hospital, Cangzhou, Hebei, China

* 18103276691@163.com

## Abstract

### Background

Characterized by high recurrence rates and limited therapeutic options, hepatocellular carcinoma (HCC) is a leading cause of cancer-related mortality worldwide. Notwithstanding the fact that telomerase-related long non-coding RNAs (TRLs) have been implicated in tumorigenesis, it remains poorly understood about their prognostic and immunological roles in HCC.

### Methods

For the purpose of identifying telomerase-related genes (TRGs) and TRLs, we used transcriptomic data from The Cancer Genome Atlas (TCGA). We built a prognostic signature using LASSO-Cox regression. Then, we validated it with time-dependent ROC curves. We assessed the model's clinical utility with nomogram calibration and DCA. We also evaluated immune profiling, tumor mutation burden, drug sensitivity and TIDE scores to characterize the tumor microenvironment. Using a pilot cohort of clinical samples, initial experimental validation was completed with RT-qPCR.

### Results

By using a 4-TRLs signature, HCC patients can be divided into Low-risk (L-R) and High-risk (H-R) groups. The signature acted as an independent prognostic factor. It provided a highly accurate prediction of patient survival at 1, 3, and 5 years (AUC: 0.744–0.770). H-R patients had more immune cells in their tumors. They also showed higher levels of checkpoint expression. Besides, their TIDE (tumor immune dysfunction and exclusion) scores were also higher. All these things mean they might not respond well to immunotherapy. Subtype-specific therapeutic vulnerabilities can be read from drug sensitivity analysis. By carrying out reverse transcription quantitative

**Data availability statement:** All relevant data are within the paper and its Supporting Information files. The public data analyzed in this study are available from the TCGA database and GEO database (accession number GSE242889). The data underlying this study have been uploaded to Zenodo and are accessible using the following: https://doi.org/10.5281/zenodo.17752944.

**Funding:** The author(s) received no specific funding for this work.

**Competing interests:** The authors have declared that no competing interests exist.

polymerase chain reaction (RT-qPCR), consistent dysregulation patterns of TRLs can be observed in HCC tissues. This providing basis supports for our bioinformatic findings. Mechanistically, lncRNA AC026356.1 linked to a telomerase-related ceRNA network. This network includes miR-126-5p and its downstream targets.

## Conclusion

The 4-TRLs signature is a tool that can be applied in HCC clinical practice. It enables prognostic stratification and helps guide treatment. These lncRNAs are linked to both immune activity and drug response. This dual role shows they affect tumor progression and the microenvironment. This finding provides new insights for precision oncology in HCC.

## Introduction

Hepatocellular carcinoma (HCC) is the main type of primary liver cancer. It makes up about 75–85% of cases worldwide [1]. HCC does not occur equally around the world. Areas in Asia and Africa report the highest number of cases. In these areas, In these areas, the disease's highest burdens have become progressively obvious [2]. HCC often has a poor prognosis because it's often found at advanced stages. Treatments then are usually less effective. The late diagnosis happens because there are no clear early signs and few trustworthy biomarkers for early detection. Recently, HCC treatment has improved with new targeted drugs and immune therapies [3,4]. Doctors use drugs like sorafenib and lenvatinib for advanced cases, which help patients live longer or better [5,6]. Similarly, immune checkpoint inhibitors that target PD-1/PD-L1 have shown promising results in clinical studies. They now offer new treatment options for some patients [7]. But even with these drugs, many patients don't live as long as we'd hope. This shows we really need special markers that can tell us how a patient can improve the condition and help doctors pick the best treatment. Thus, making trustworthy tools to predict outcomes is super important. It'll help us give each patient care that fits them best and improve their chances of surviving HCC.

Telomerase is a special mix that makes and protects the end parts of chromosomes [8]. Telomeres are made up of repeated DNA bits (like TTAGGG in people). They guard chromosomes and keep our genes stable [9]. Telomerase makes telomeres longer, balancing out the shortening that happens when cells divide. This slows down cell aging and keeps cells dividing for longer [10]. This enzyme works in some normal body parts (like germ and stem cells) and gets turned back on in cancer cells. It enables them to copy themselves endlessly, which is a sign of cancer [11]. Besides affecting cells itself, telomerase also changes the tumor's immune environment. For instance, it's connected to more immune-blocking cells, like regulatory T cells. The above effects are helpful for tumors to evade immune destruction [12].

For transcripts longer than 200 nucleotides, they can be reckoned as long non-coding RNAs (lncRNAs). They do not encode proteins. Although they lack coding potential, lncRNAs can regulate gene expression. To go it further, they can cause

diverse biological processes [13]. Their involvement in cancer development, progression, and patient prognosis across tumor types has gained validation in many researches [14–17]. Moreover, lncRNAs are now recognized as key modulators of antitumor immunity, influencing immune recognition, cell recruitment, and effector functions. [18]. Nonetheless, it's still ambiguous about the roles of telomerase-related lncRNAs (TRLs) in HCC. To assess its ability to predict survival and inform treatment strategies in HCC, this study is intended to address this gap by developing a TRLs-based prognostic model.

## Materials and methods

### Data acquisition and processing

TCGA database (https://portal.gdc.cancer.gov/) offered transcriptomic data, somatic mutation profiles, and corresponding clinical information for HCC. The above data, involved 374 tumor specimens and 50 adjacent normal liver tissues. Through the utilization of Perl (strawberry-perl-5.30; https://www.perl.org), raw sequencing data were processed properly. The purpose was to generate standardized mRNA and lncRNA expression matrices, calculate tumor mutation burden (TMB), and compile survival annotations. TRGs were systematically curated from the GeneCards database (https://www.genecards.org, S1 Table). Single-cell RNA sequencing data were obtained from the GEO database under accession number GSE242889 (https://www.ncbi.nlm.nih.gov/geo/query/acc.cgi?acc=GSE242889). From this dataset, we selected five primary hepatocellular carcinoma samples (GSM7774399, GSM7774400, GSM7774401, GSM7774402, and GSM7774403) for subsequent analysis. Additionally, six independent external validation datasets were obtained from the GEO database. These datasets included: GSE14520 (225 HCC and 220 normal samples), GSE121248 (70 HCC and 37 normal samples), GSE76427 (115 HCC and 52 normal samples), GSE174570 (57 HCC and 57 paired normal samples), GSE54236 (81 HCC and 80 normal samples), and GSE17856 (225 HCC and 52 normal samples).

### Differential expression analysis of TRGs

Through systematic interrogation of the GeneCards database, we curated a refined set of 2,867 protein-coding TRGs. Expression matrices of these TRGs were subsequently extracted from the combined TCGA cohort by employing the 'limma' R package. Differential expression analysis between HCC and normal tissues was performed in the TCGA-LIHC by utilizing the 'limma' package. Statistical significance thresholds were set at an absolute $\log_2$ fold-change >1 and adjusted P-value <0.001. Final visualization of differentially expressed genes (DEGs) was achieved through volcano plots and hierarchical clustering heatmaps generated with the 'ggplot2' and 'pheatmap' packages, severally.

### Identification of key TRGs-associated modules via WGCNA

Weighted gene co-expression network analysis (WGCNA) was applied to delineate HCC-associated modules within the TRGs expression profile. An optimal soft-thresholding power (β = 17) was selected to ensure a scale-free network topology (scale-free topology fit index $R^2 > 0.9$). Module identification was conducted by adopting hierarchical clustering with a minimum module size of 50 genes and a merge cut height of 0.25. Genes significance and module membership correlations were systematically evaluated between HCC and normal tissue samples. The most differentially expressed modules can be precisely identified by this analysis. Moreover, they should be a priority for further investigation.

### Protein-protein interaction and functional enrichment analyzes of TRGs

Through using the STRING database (version 11.5; https://string-db.org/) with species-specific parameters, protein-protein interaction (PPI) analysis of candidate TRGs was performed for Homo sapiens. For the purpose of defining significant interactions, a medium confidence interaction score threshold of >0.4 was applied correspondingly. By using the

 

cytoHubba plugin within Cytoscape software version 3.10.1), we performed network analysis to refine the selection of pivotal TRGs. Within the protein-protein interaction network, systematic evaluation and prioritization of key regulatory genes can be materialized through this approach. This methodology was based on multiple topological algorithms. They normally encompassed Degree, Edge Percolated Component (EPC), Maximum Neighborhood Component (MNC), and Maximal Clique Centrality (MCC). In the TRG interaction network, the most influential nodes are the resulting hub genes. Moreover, they could be employed for subsequent functional validation.

It is particularly noteworthy that the Molecular Signatures Database (MSigDB; http://software.broadinstitute.org/gsea/msigdb/) was the source of gene sets from the Kyoto Encyclopedia of Genes and Genomes (KEGG) database (c5.go.v7.4.symbols.gmt and c2.cp.kegg.v7.4.symbols.gmt). Through utilizing the clusterProfiler R package (version 4.4.2), functional enrichment analysis of candidate genes was conducted systematically. It aimed to identify significantly enriched Gene Ontology (GO) terms. The discussed terms normally included biological processes (BP), molecular functions (MF), and cellular components (CC), as well as KEGG pathways. Terms and pathways were considered statistically significant at a false discovery rate (FDR)-adjusted P-value <0.05 using the Benjamini–Hochberg correction method.

## Construction and validation of the prognostic predictive model

Co-expression analysis between hub TRGs and lncRNA expression matrices was performed using stringent criteria (Pearson correlation coefficient >0.4, P < 0.001). Resultant co-expression relationships were visualized using Sankey diagrams generated with the "ggalluvial" R package. Expression profiles of TRLs were integrated with clinical data from HCC patients using the "limma" R package to generate a unified dataset. The dataset was randomly partitioned into training and testing sets using the createDataPartition function from the caret package (version 7.0.1). To ensure reproducibility, a fixed random seed of 222 was set prior to partitioning. Stratified sampling was performed based on the survival status to maintain similar outcome distributions in both sets. The data were split into 50% training and 50% testing subsets (p = 0.5). Through LASSO regression, a prognostic model was developed in the training cohort, followed by multivariate Cox proportional hazards regression analysis. The LASSO regression was performed using the "glmnet" package with ten-fold cross-validation, where the optimal penalty parameter (λ) was selected based on the minimum cross-validation error criterion (λ.min). By using the "survival" package, multivariate Cox analysis was conducted subsequently to optimize the feature selection. The model performance was evaluated by employing the "survminer" and "timeROC" packages. The final model was subsequently validated in both the testing cohort and the entire dataset. Individual risk scores were calculated by adopting the linear predictor derived from the multivariate Cox model: (x(i)) sourced from the multivariate Cox regression analysis as well as the associated coefficients of TRLs (coef(i))

$$Risk Score = \sum_{i=1}^{n} coef(i) * x(i)$$

Patients in the training, testing, and full datasets were stratified into high-risk (H-R) and low-risk (L-R) groups based on the median risk score. Kaplan–Meier survival curves were generated by utilizing the "survival" package to compare OS between risk groups. Time-dependent receiver operating characteristic (ROC) curves were plotted and area under the curve (AUC) values were computed by employing the "survival," "survminer," and "timeROC" packages to evaluate the predictive accuracy of the model. Through the utilization of the R packages "kernelshap" and "shapviz," SHAP (SHapley Additive exPlanations) analysis was conducted accordingly. The purpose was to enhance the interpretability of the 4-TRLs prognostic model. The contribution of each feature to individual predictions can be quantified when this approach was adopted. Hence, an integrated assessment of both global model behavior and instance-specific decision patterns can be realized by adopting this method.

## Construction and validation of a prognostic nomogram

By employing the training cohort data, this research developed a comprehensive nomogram to predict OS in HCC patients at 1, 3, and 5 years. To be specific, this nomogram integrated the 4-TRLs signature and key clinical variables. Through the utilization of the "rms," "survival," "regplot," and supporting packages ("dplyr," "pec," "Survcomp"), model construction was performed for validating and visualizing the results. Through concordance index (C-index) calculation and calibration curve analysis, the predictive accuracy of the nomogram underwent quantitative evaluation. In comparison with alternative strategies, DCA was further employed to evaluate the clinical utility and net benefit of the nomogram for 1-, 3-, and 5-year survival prediction.

## Analysis of tumor immune cell infiltration and immune function

To characterize the immune landscape of HCC patients, the CIBERSORT algorithm was employed correspondingly. In an effort to systematically compare the infiltration levels of 22 immune cell types between the H-R and L-R cohorts, a suite of R packages comprising "CIBERSORT," "preprocessCore," "parallel," "bseqsc," and "e1071" was employed in the aftermath of crucial steps. Additionally, the robustness of the immune infiltration estimates was validated using multiple independent algorithms, comprising XCELL, TIMER, QUANTISEO, MCPCOUNTER, EPIC, and CIBERSORT-ABS. On top of that, correlations between the 4-TRLs signature and 13 immunoregulatory pathways were evaluated by utilizing the "limma," "GSVA," "GSEAbase," "ggpubr," and "reshape2" packages [19,20]. Through the utilization of the "limma," "reshape2," "ggplot2," and "ggpubr" packages, differential expression of immune checkpoint genes across risk cohorts underwent systematic assessment. Finally, by adopting the "RColorBrewer" package, an immune subtype map was generated for TCGA-LIHC samples. The main purpose was to delineate two things. One is the distributions of immune subtypes. The other is their association with clinical outcomes.

## Drug sensitivity analysis

Leveraging publicly available data from the Genomics of Drug Sensitivity in Cancer (GDSC) database (https://www.cancerrxgene.org/), we performed drug sensitivity profiling by adopting ridge regression implemented in the "pRRophetic" R package. In both H-R and L-R HCC cohorts, this approach enabled estimation of half-maximal inhibitory concentration (IC$_{50}$) values for multiple anti-tumor agents. To evaluate differential drug sensitivity patterns across risk strata and therapeutic compounds, comparative analysis of IC$_{50}$ values was conducted subsequently.

## Consensus clustering analysis

Through the utilization of the R package "ConsensusClusterPlus," this research performed unsupervised consensus clustering of TCGA-LIHC samples. Such method was rooted in the quantitative profiles of 22 immune cell subtypes derived from the CIBERSORT algorithm. In line with distinct immune infiltration patterns, robust molecular stratification of HCC specimens can be materialized via this approach. With an aim to evaluate differences in immune cell infiltration and immune checkpoint gene expression across the identified clusters, subsequent comparative analyzes were conducted by utilizing the "reshape2," "ggpubr," "limma," and "ggplot2" packages. The purposes of these analyzes were diverse. One of the paramount was to characterize immune microenvironment heterogeneity among molecular subtypes of hepatocellular carcinoma.

## Assessment of tumor mutational burden and immunotherapy response

Body cell change information for HCC was got from TCGA. We dealt with it using Perl. We used the "maftools" R package to get the TMB values. Then, we checked if TMB was related to how long patients lived in different risk groups. We obtained pan-cancer Stem Cell (CSC) scores from the UCSC Xena database (http://xena.ucsc.edu/). After that, their

correlation with the risk cohorts was investigated by employing R packages "ggpubr," "ggplot2," "ggExtra," and "limma." Additionally, potential differences in immunotherapy response between risk groups were predicted by utilizing the TIDE computational platform (http://tide.dfci.harvard.edu/).

### Patient recruitment and sample collection

Between January 2025 and September 2025, a total of 15 patients with HCC were enrolled from Cangzhou People's Hospital. All participants provided written informed consent prior to inclusion in the study. The inclusion criteria consisted of: (1) histopathologically confirmed diagnosis of HCC; (2) age 18 years or older. Exclusion criteria were: (1) history of any malignancy; (2) prior treatment for HCC; (3) pregnancy.

### RT-qPCR

Total RNA was extracted from paired hepatocellular carcinoma and adjacent non-tumor tissues by utilizing the RNA Extraction Kit (Shanghai Hifun Biotechnology Co., Ltd.; Cat. No. EZB-RN001A) in accordance with the manufacturer's instructions. Complementary DNA (cDNA) was synthesized by employing the 4×EZscript Reverse Transcription Mix II (with gDNA Remover) (Shanghai Hifun Biotechnology Co., Ltd.; Cat. No. EZB-RT2GQ), following the supplier's recommended protocol.

Quantitative PCR was performed by adopting a real-time fluorescence detection system and the 2×SYBR Green qPCR Master Mix (ROX2) (Shanghai Hifun Biotechnology Co., Ltd.; Cat. No. A0001-R2). Each 20 μL reaction mixture contained 10 μL of Master Mix, 0.2 μL of each forward and reverse primer, 2 μL of cDNA template, and 7.6 μL of nuclease-free water. The amplification conditions consisted of an initial denaturation at 95°C for 5 min, followed by 40 cycles of denaturation at 95°C for 10 sec, and annealing/extension at 60°C for 30 sec. A melting curve analysis was subsequently performed to confirm amplification specificity.

The relative expression level of the target gene was calculated. We used the $2^{-\Delta\Delta Cq}$ method for this calculation. The GAPDH gene was used as a reference to get the result [21,22]. We checked how tumor tissues and nearby normal tissues are different. A test called Student's t-test was used for this. We looked at the two-sided p-value. If this value is less than 0.05, it is considered statistically important. S2 Table presents primer sequences for the target and reference genes.

### Statistical analysis

Statistical analyzes were performed using GraphPad Prism (version 9.3.0), Perl (version 5.30.0), and R (version 4.4.2). Methodological approaches included LASSO regression for feature selection, principal component analysis (PCA) for dimensionality reduction, ROC curve analysis for predictive accuracy evaluation, Kaplan-Meier survival analysis with log-rank testing, and univariate and multivariate Cox proportional hazards regression modeling. Gene expression data from TCGA (mRNA and lncRNA) were procured in FPKM format and log2-transformed for analysis; single-cell data from GEO were normalized via Seurat's LogNormalize method and scaled to z-scores. Regarding missing clinical data, we implemented context-specific handling strategies: for correlation analyzes, all samples comprising those with "unknown" entries were retained; for prognostic model construction, all available HCC samples (N=370) were included regardless of missing clinical annotations; for subgroup survival analyzes, samples with missing clinical features were excluded to maintain well-defined clinical categories. For RT-qPCR data, gene expression levels were normalized using the $2^{-\Delta\Delta Cq}$ method. Differences in expression between hepatocellular carcinoma tissues and adjacent non-tumor tissues were assessed using two-tailed Student's t-tests, with a p-value less than 0.05 considered statistically significant.

### Ethics approval

The Ethics Committee at Cangzhou People's Hospital approved this study. Permission was also given in writing by all participants before any samples were taken.

## Results

### Identification of differentially expressed genes and WGCNA

Fig 1 illustrates what kinds of methodological framework this study employed. To get the TRGs we needed, we used the "limma" package in R. This helped us pull out 2835 genes from the TCGA data. We checked how genes are different in HCC tissues and normal tissues. A tool called "limma" was used for this. And we followed strict rules, checked how genes are different in HCC tissues and normal tissues. A tool called "limma" was used for this. And we followed strict rules (logFC > 1 and adj. P. Val < 0.001). From the TCGA data, 435 genes that were different were identified. There were 276 up-regulated ones and 159 down-regulated ones. These are revealed in a volcano plot (Fig 2A). The up genes are marked in red, and the down genes are in blue. Fig 2B displayed the first 60 DEGs on the heat map. WGCNA identified gene modules correlated with HCC phenotypes. Soft-thresholding power (β) of 17 was implemented to achieve scale-free topology (R² > 0.9; Fig 2C). This power value facilitated construction of adjacency and topological overlap matrices (TOM). Hierarchical clustering based on TOM-based dissimilarity revealed four distinct co-expression modules (Fig 2D). In contrast to others, the blue module exhibited the strongest HCC association, demonstrating more favorable gene significance (Fig. 2E, r = 0.48, p = 5e-26). Figs 2F and 2G separately demonstrate the gene distribution of blue module in tumor and normal tissues. Integration of TRGs from the DEGs and blue module genes yielded 141 candidate TRGs (Fig 2H).

### PPI and functional enrichment analyzes of candidate TRGs

PPI analysis was performed by employing the STRING database with a confidence interaction score threshold >0.4. Fig 3A depicts the resulting PPI network. Subsequent network analysis via Cytoscape identified 16 hub TRGs most critically associated with HCC pathogenesis: CCNA2, CDK1, CDC20, CCNB1, BUB1B, KIF20A, BUB1, TOP2A, CENPF, PLK1, UBE2C, BIRC5, MAD2L1, CEP55, CENPE, and CENPA (Fig 3B).

Functional enrichment analysis of the 141 candidate TRGs revealed significant terms and pathways. Within GO BP, the top three enriched terms were regulation of cell cycle phase transition, nuclear division, and mitotic cell cycle phase transition. With regard to GO CC, the most enriched terms were chromosomal region, condensed chromosome, and chromosome, telomeric region. Under GO MF. Nonetheless, the leading categories were catalytic activity, exerting certain influences on DNA, DNA helicase activity, and single−stranded DNA helicase activity (Figs 3C–D). KEGG pathway analysis highlighted Cell cycle, DNA replication, and Cellular senescence as the most strikingly enriched pathways (Figs 3E–F). Further enrichment analysis revealed the top 10 enriched terms in GO BP, CC, and MF, as well as KEGG pathways for the 16 hub TRGs (S1 Fig).

### Prognostic signature construction and validation

First, 16876 lncRNAs, coupled with hub TRGs were extracted from the TCGA database for co-expression analysis. By adopting |R| > 0.4 and p < 0.001 as the analysis criteria, 424 lncRNAs and 16 candidate TRGs were finally identified for co-expression, and visualization was achieved by utilizing Sankey diagrams (Fig 4A, S3 Table).

In TCGA-LIHC dataset, 374 tumor samples were initially identified. Subsequent to the exclusion of four cases with incomplete survival data, 370 patients were included in the final cohort and randomly allocated into training (n = 185) and validation (n = 185) sets at a 1:1 ratio. The training set was utilized to identify prognostic TRLs and to construct the predictive model, while the test set was reserved for assessing model accuracy. These two cohorts demonstrated well-balanced baseline clinical characteristics, comprising age, gender, tumor grade, stage, and TNM classification, with no inter-group statistically significant differences (Table 1; all p > 0.05). Univariable Cox regression analysis identified 175 TRLs linked to HCC prognosis. All of these TRLs were associated with higher risk, as shown in S4 Table.. To reduce multi-collinearity and overfitting, we first applied LASSO regression to the 175 prognostic TRLs. Then, using the optimal lambda value, we pinpointed 11 TRLs that remained strongly linked to HCC patient outcomes (Figs. 4B, C). Multivariable Cox regression

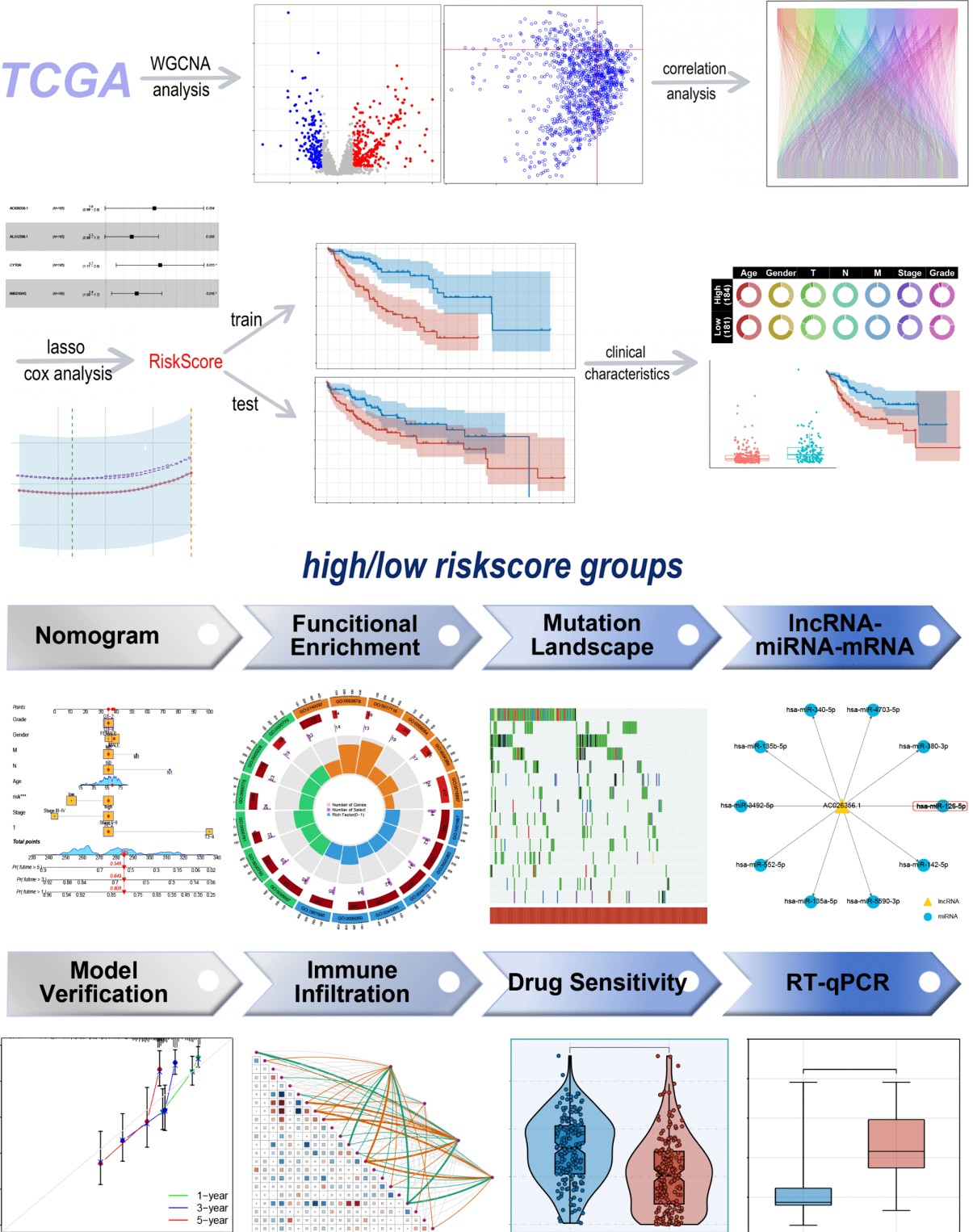

**Fig 1. Study workflow diagram.** Workflow diagrams are utilized to describe the entire research process.

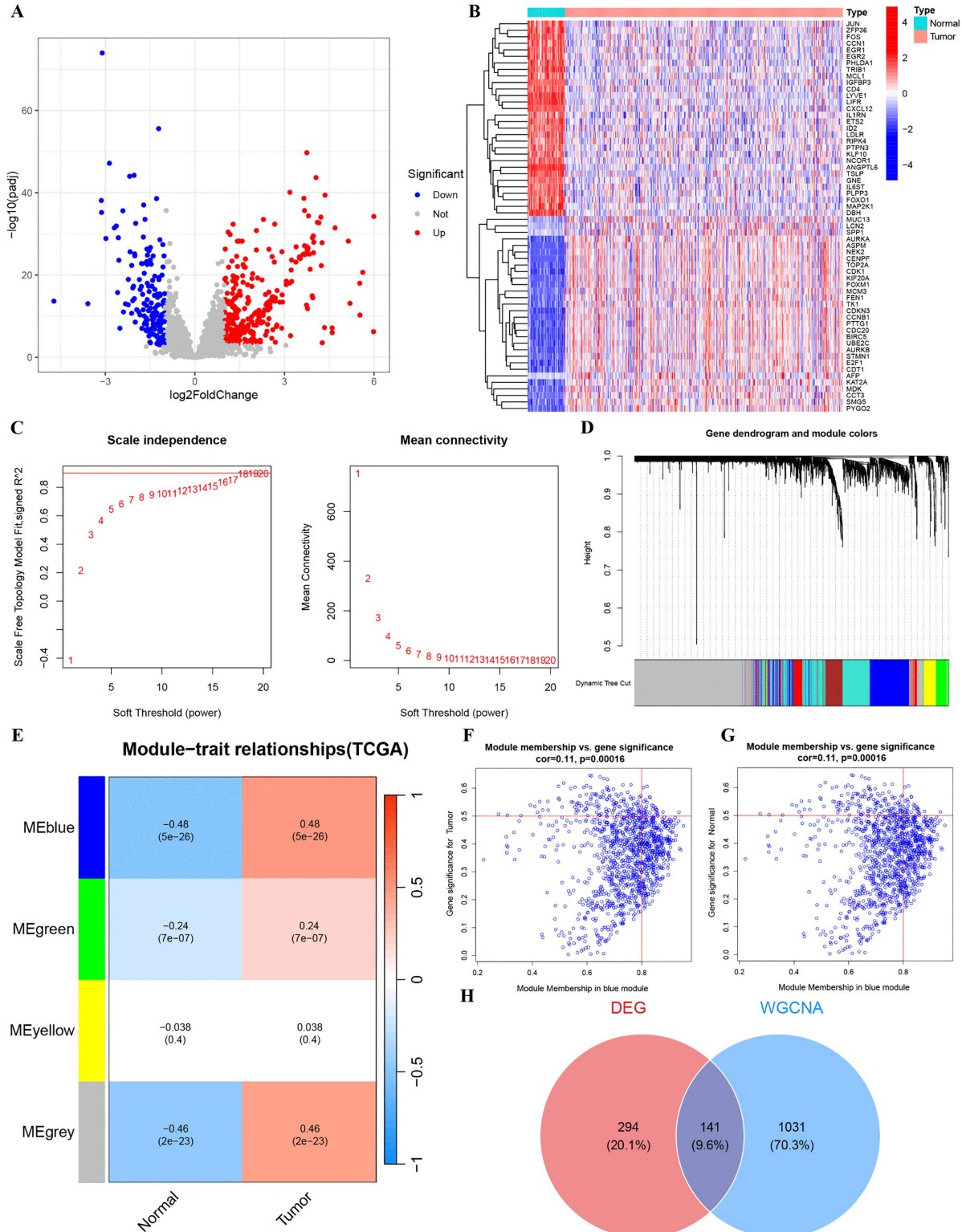

**Fig 2. Identification of DEGs and WGCNA.** (A) Volcano plot illustrating differential expression of TRGs between HCC and normal tissues in the TCGA-LIHC. (B) The heat map suggests the top 60 DEGs. (C) Selection of the optimal soft-thresholding power for network construction. (D) Identification of gene co-expression modules through dynamic tree cutting. (E) Heatmap depicting the correlation between module eigengenes and clinical traits. (F, G)

Scatter plots of module membership against gene significance for the blue module in HCC samples (F) and normal samples (G). (H) The Venn diagram suggests 141 candidate TRGs.

analysis was subsequently performed, ultimately selecting four TRLs for incorporation into the final prognostic prediction model (Fig 4E, S5 Table). The 4-TRL-based risk score was calculated in accordance with the following formula:

Riskscore=exprAC026356.1 × 0.478060335762901 + exprAL512598.1 × 0.254946101377735 + exprCYTOR×0.534456923673206 + exprMIR210HG×0.303996696168745, each patient's risk score was further calculated. For each TRL in the formula, Fig 4D suggests the correlation coefficients. Detailed summary of the screening procedures and criteria for TRGs and TRLs is provided in Supplementary Table S6. Rooted in median risk score, the TCGA-LIHC cohort can be divided into H-R and L-R cohorts. As the results displayed, in the H-R group, AC026356.1, AL512598.1, CYTOR and MIR210HG had much higher amounts. A positive number was observed in their math formula for these findings. This means there could be reasons that make the risk go up. We applied SHAP analysis to quantify the contribution of each feature to patient outcomes after building the prognostic model. The SHAP summary plot (Fig 4F) ranks the four TRLs by their mean absolute SHAP value, which reflects each feature's overall contribution to the model's predictions. In the list of influential features (mean |SHAP| = 0.268), MIR210HG was put at the top. The next factors were CYTOR (0.254), AC026356.1 (0.199), and AL512598.1 (0.186), separately. In prognostic risk stratification, the relative importance of each TRLs can be more conspicuous under this kind of order. Fig 4G is a beeswarm graph. It shows how each thing's SHAP values are spread out in all the samples. And each dot stands for one single sample. On the graph, yellow is used for high numbers and purple for low ones. If the dots are mostly on the right, it means something bad. If they are on the left, it means something good. Where the dots are crowded shows the most common result. The SHAP waterfall plot (Fig 4H) and force plot (Fig 4I) visualize how each feature shifts the model output from the baseline expected value (E[f(x)] = 1.67) toward the final prediction (f(x) = 1.69). For instance, CYTOR contributed +0.567, while AC026356.1 contributed −0.309. This plot also suggests threshold effects for individual variables, suggesting that exceeding certain values may enhance predictive performance. In addition, to test the prognostic performance of the 4-TRLs signature, the 1-year, 3-year and 5-year AUC values of 4-TRLs were compared with 16 other published prognostic signatures in HCC patients in the TCGA database, which exhibited associations with some biological features, comprising cuproptosis, hypoxia, metabolism, etc. The analysis demonstrated that the 4-TRLs signature exhibited superior predictive accuracy for 3-year (AUC = 0.759) and 5-year (AUC = 0.770) survival outcomes compared with most existing models, though it exhibited relatively limited discriminative ability for 1-year (AUC = 0.744) survival prediction (S2 Fig, S7 Table).

## Clinical characteristics and survival analysis

Within the TCGA-LIHC cohort, the 4-TRLs signature exhibited significant associations with key clinicopathological features, as visualized in a circos plot (Fig 5A). In a heat map, the associations between H-R and L-R groups are more conspicuous. Various clinical characteristics are also further validated (Fig 5B). In accordance with vital status (Alive vs Dead; Fig 5C), pathological grade (grade 1 vs 2, 1 vs 3, 1 vs 4, and 2 vs 3, Fig 5F), tumor stage (stage I vs II and I vs III; Fig 5G), and T category (T1 vs T2, T1 vs T3, and T1 vs T4; Fig 5H), statistically significant differences (P < 0.05) in 4-TRLs expression can be highlighted by stratified analyses. Nonetheless, age, gender, N stage, and M stage didn't show any clear links to the 4-TRLs signature (Figs 5D, E, and Figs 6I, J; P > 0.05).

We performed subgroup survival analyzes across age, gender, grade, stage, and TNM categories to evaluate the prognostic relevance of the 4-TRLs signature in diverse clinical contexts. Results consistently demonstrated that patients in the H-R group had significantly poorer overall survival compared to those in the L-R group across all subgroups (S3 Fig), suggesting its potential applicability for prognostic stratification in these specific subgroups of HCC patients.

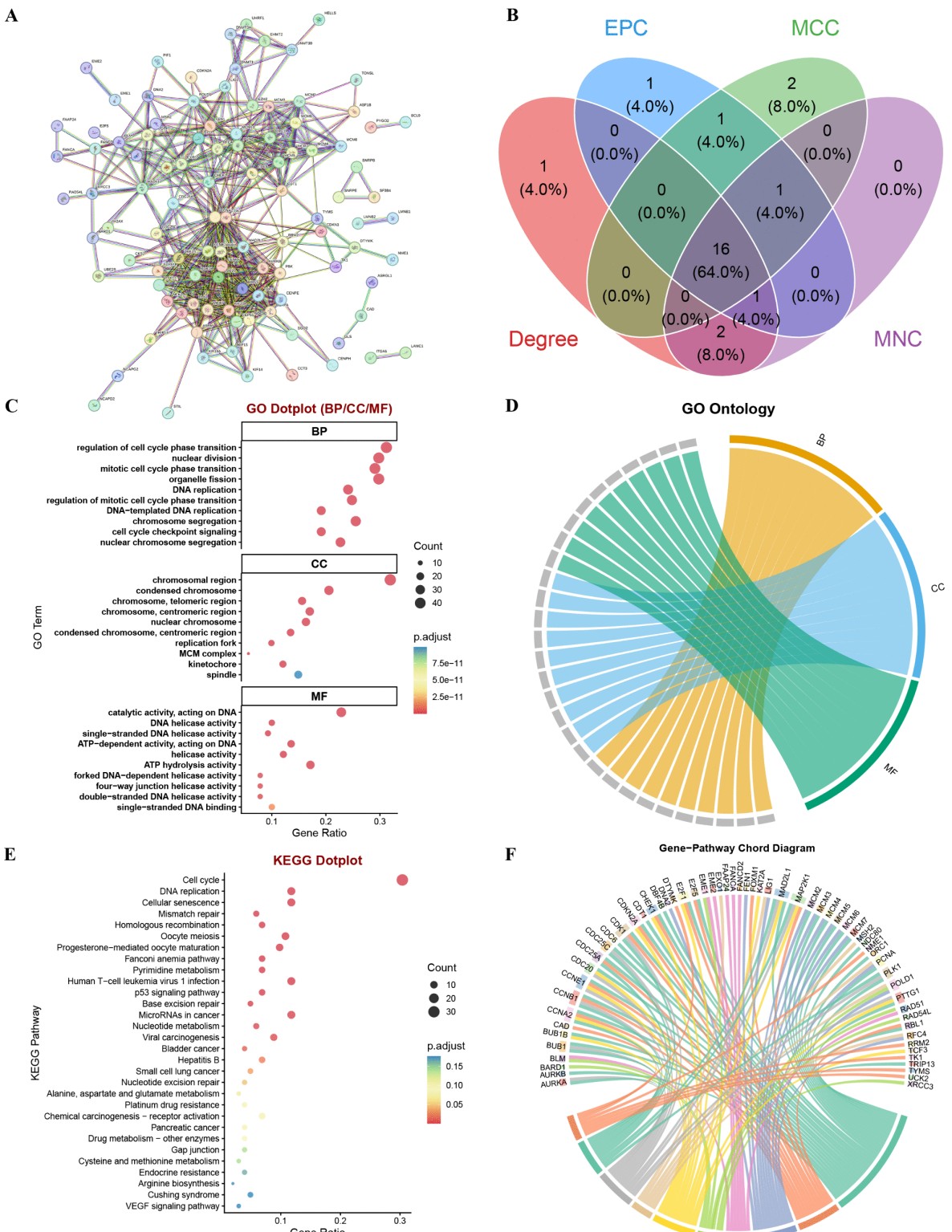

**Fig 3. PPI and functional enrichment analyzes of candidate TRGs.** (A) PPI network of the 141 candidate TRGs derived from the STRING database, with an interaction score threshold > 0.4. (B) The Venn diagram suggests the further screening of hub TRGs through four algorithms: Degree, EPC, MCC, and MNC. (C) Bubble plot denotes GO enrichment analysis. (D) Chord diagram visualizes associations in the GO enrichment analysis. (E) Bubble plot displays KEGG pathway enrichment results. (F) Chord diagram illustrates KEGG pathway associations.

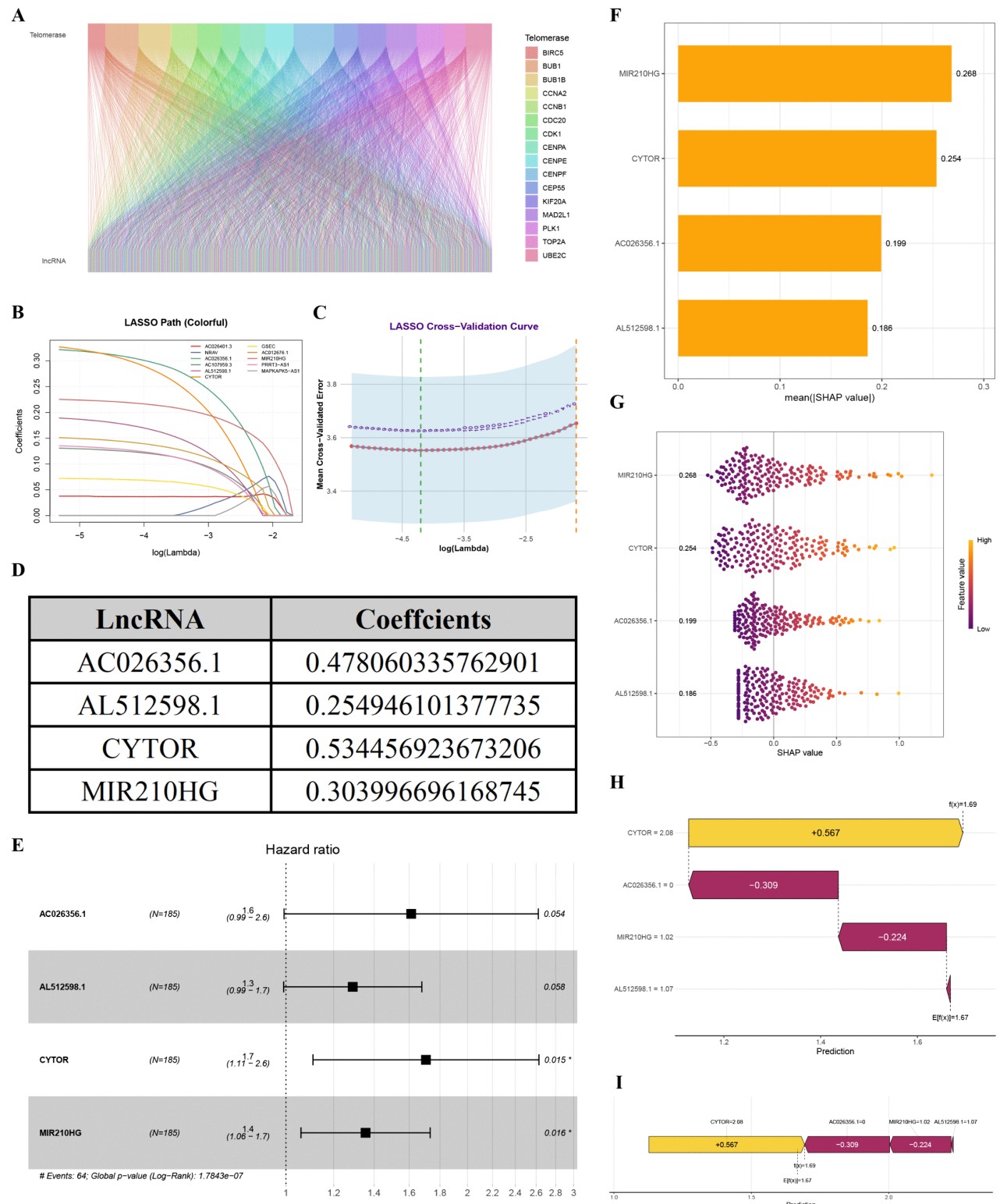

**Fig 4. Prognostic signature construction and validation.** (A) Sankey diagram illustrating co-expression associations between TRGs and lncRNAs. (B) Further screening of prognostic TRLs via LASSO regression analysis. (C) Cross-validation of LASSO regression. (D) Four TRLs and their respective correlation coefficients. (E) Four TRLs were selected through multivariate COX regression analysis. (F) Summary plot of SHAP values illustrates the

overall contribution of each feature to model predictions. (G) Beeswarm plot exhibits the distribution and direction of SHAP values for individual features across the cohort. (H, I) Waterfall plot (H) and force plot (I) visualize the additive contribution of each feature to a specific prediction, demonstrating how the base value is adjusted to the final output.

**Table 1. Homogeneity of the training and testing sets in terms of the baseline characteristics of clinical indicators.**

| Variates | Type | Training set (n = 185) | | Testing set (n = 185) | | All set (n = 370) | | P-value |
|---|---|---|---|---|---|---|---|---|
| Age | ≤ 65 | 115 | 62.16% | 117 | 63.24% | 232 | 62.7% | 0.914 |
| | > 65 | 70 | 37.84% | 68 | 36.76% | 138 | 37.3% | |
| Gender | FEMALE | 65 | 35.14% | 56 | 30.27% | 121 | 32.7% | 0.375 |
| | MALE | 120 | 64.86% | 129 | 69.73% | 249 | 67.3% | |
| Grade | G1 | 29 | 15.68% | 26 | 14.05% | 55 | 14.86% | 0.696 |
| | G2 | 92 | 49.73% | 85 | 45.95% | 177 | 47.84% | |
| | G3 | 56 | 30.27% | 65 | 35.14% | 121 | 32.7% | |
| | G4 | 5 | 2.70% | 7 | 3.78% | 12 | 3.24% | |
| | unknow | 3 | 1.62% | 2 | 1.08% | 5 | 1.35% | |
| Stage | Stage I | 93 | 50.27% | 78 | 42.16% | 171 | 46.22% | 0.235 |
| | Stage II | 35 | 18.92% | 50 | 27.03% | 85 | 22.97% | |
| | Stage III | 41 | 22.16% | 44 | 23.78% | 85 | 22.97% | |
| | Stage IV | 3 | 1.62% | 2 | 1.08% | 5 | 1.35% | |
| | unknow | 13 | 7.03% | 11 | 5.95% | 24 | 6.49% | |
| T | T1 | 98 | 52.97% | 83 | 44.86% | 181 | 48.92% | 0.288 |
| | T2 | 39 | 21.08% | 54 | 29.19% | 93 | 25.14% | |
| | T3 | 39 | 21.08% | 41 | 22.16% | 80 | 21.62% | |
| | T4 | 13 | 3.51% | 7 | 3.78% | 6 | 3.24% | |
| | unknow | 3 | 0.81% | 0 | 0% | 3 | 1.62% | |
| M | M0 | 266 | 71.89% | 137 | 74.05% | 129 | 69.73% | 0.583 |
| | M1 | 4 | 1.08% | 1 | 0.54% | 3 | 1.62% | |
| | unknow | 100 | 27.03% | 47 | 25.41% | 53 | 28.65% | |
| N | N0 | 252 | 68.11% | 124 | 67.03% | 128 | 69.19% | 0.603 |
| | N1 | 4 | 1.08% | 3 | 1.62% | 1 | 0.54% | |
| | unknow | 114 | 30.81% | 58 | 31.35% | 56 | 30.27% | |

## Assessment of the prognostic signature

A risk score distribution plot integrated with survival outcomes and expression patterns of the 4-TRLs signature revealed that patients in the H-R cohort had remarkably shorter survival times, poorer clinical outcomes, and elevated expression of risk-associated transcripts compared to those in the L-R cohort within the training set (Fig 6A). Consistent findings were observed in both the testing set and the entire cohort when employing identical analytical approaches, thereby confirming the reproducibility of the risk stratification (Figs 6B, C). Kaplan-Meier survival analysis demonstrated that patients in the H-R group had significantly worse OS compared to the L-R group in the training set (log-rank p < 0.05). This prognostic distinction was consistently validated in both the testing set and the full cohort (Figs 6D-F). ROC analysis exhibited that the AUC values for 1-, 3-, and 5-year survival prediction were 0.744, 0.759, and 0.770, severally, in the training set; 0.683, 0.614, and 0.584 in the testing set; and 0.712, 0.678, and 0.676 in the entire cohort (Figs 6G-I). Aside from that, when we pitted the 4-TRLs signature against regular clinicopathological parameters, the clinical ROC curves showed it's better

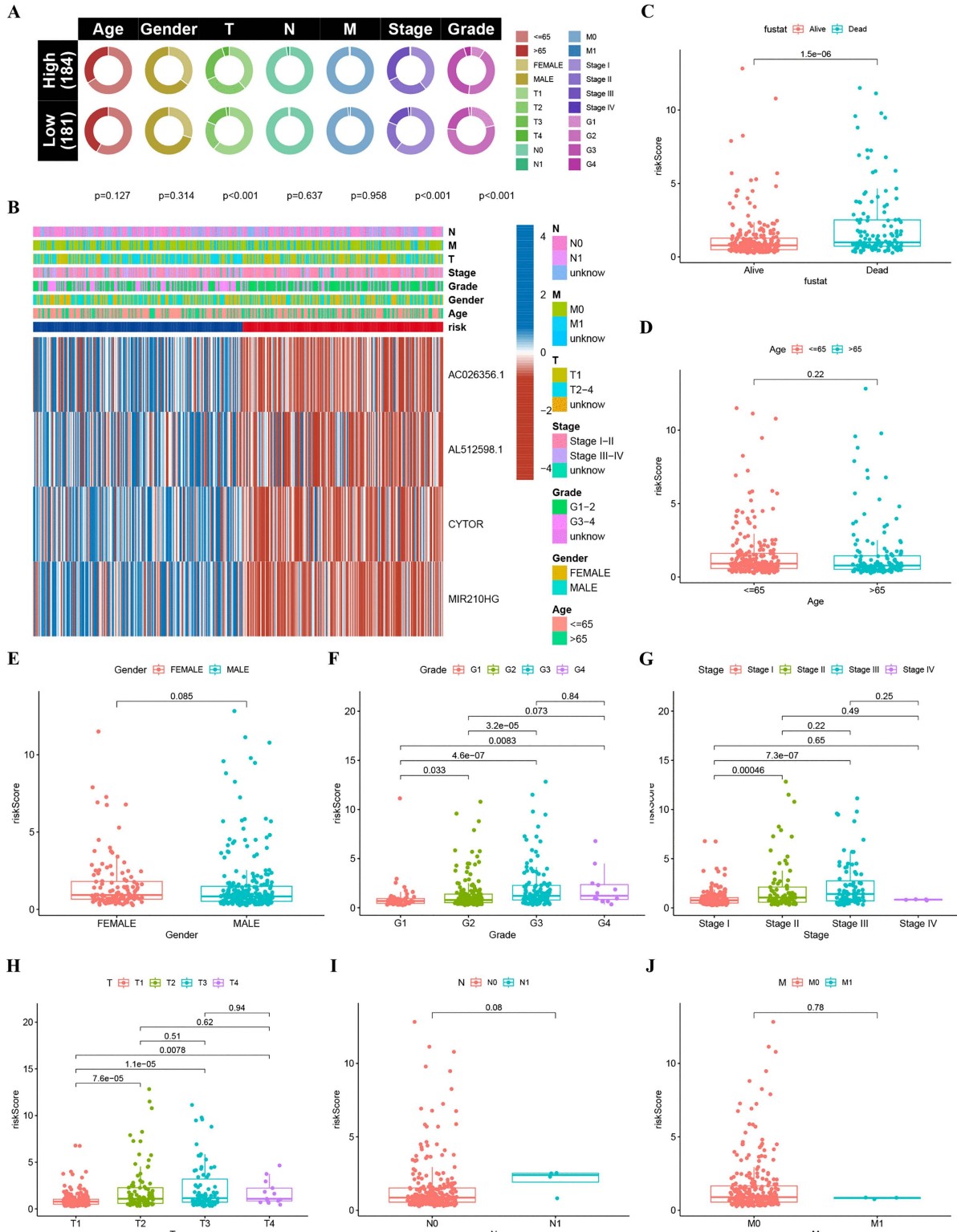

**Fig 5. Clinical characteristics analysis.** (A) Circos plot of clinicopathological characteristics and comparative distribution between L-R and H-R groups. (B) Heat map visualizing the correlation between two risk groups and clinical characteristics. (C-J) Boxplot of 4-TRLs rooted in TRLs signature in HCC patients with different fusat (C), ages (D), genders (E), pathological grades (F), tumor stages (G), T stages (H), N stages (I) and M stages (J).

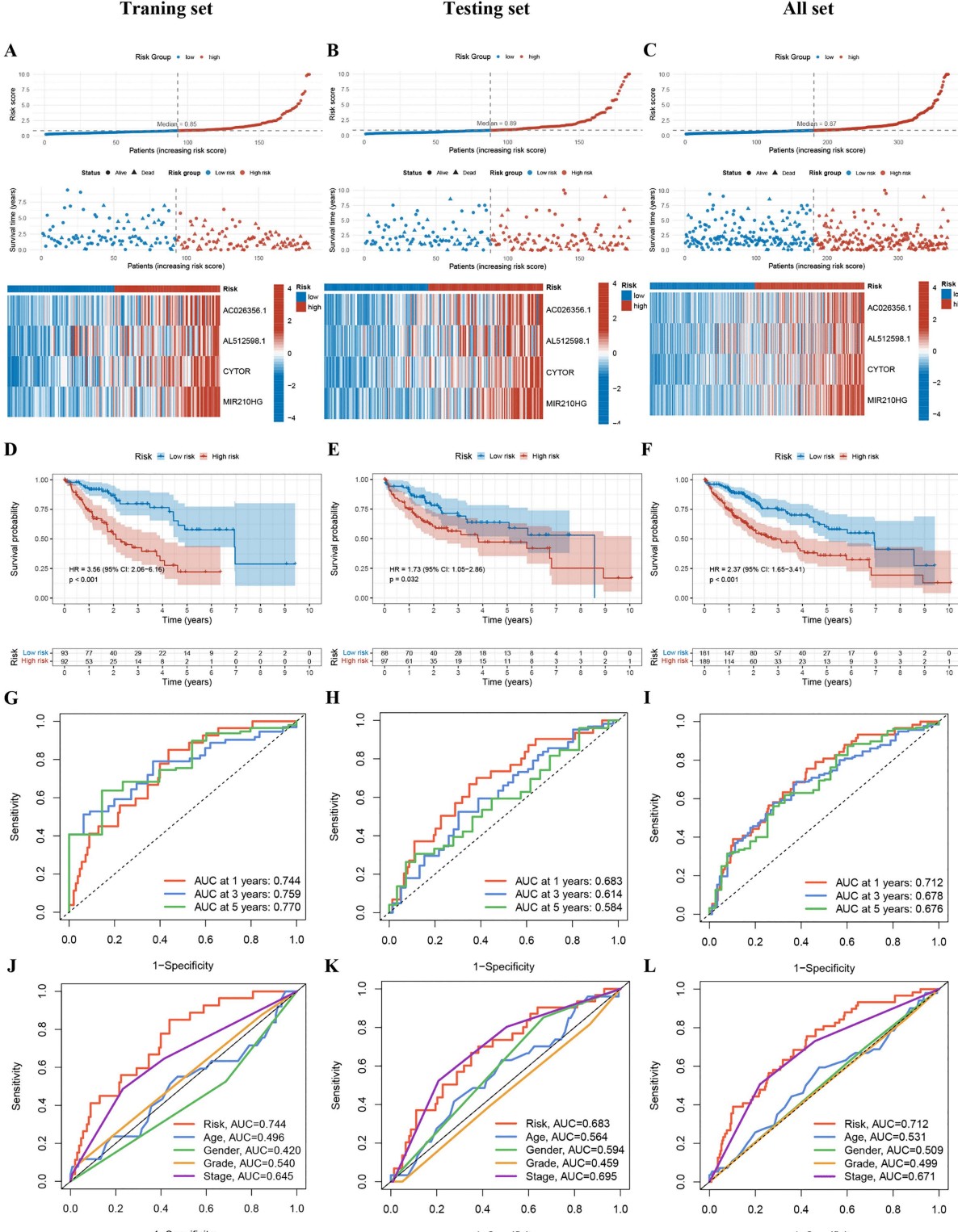

**Fig 6. Assessment of the prognostic signature.** (A-C) Distribution of risk scores grounded in the 4-TRLs signature, survival time, survival status, and expression heatmap of the four TRLs in the (A) training, (B) testing, and (C) all sets. (D-F) Kaplan-Meier survival curves comparing overall survival

between H-R and L-R groups in the (D) training, (E) testing, and (F) all sets. (G-I) ROC curves evaluating the predictive accuracy of the risk model in the (G) training, (H) testing, and (I) all sets. (J-L) ROC curves comparing the prognostic performance of the 4-TRLs signature against conventional clinical parameters in the (J) training, (K) testing, and (L) all sets.

at guessing outcomes across all datasets, with AUC values of 0.744 (training set), 0.683 (testing set), and 0.712 (all set) (Figs 6J-L). These results suggest that the 4-TRLs-based prognostic model reliably predicts outcomes in patients with hepatocellular carcinoma and suggests robust generalizability across independent cohorts.

## Nomogram construction and validation

Cox regression analysis was performed to identify independent predictors of HCC prognosis by evaluating the 4-TRLs signature alongside clinicopathological parameters within the training cohort. As illustrated in the forest plot (Fig 7A), stage III ($P < 0.001$), stage IV ($P < 0.05$), and the 4-TRLs signature ($P < 0.001$) emerged as significant independent risk factors for HCC prognosis. The results in Fig 7B show that our 4-TRLs model was better at predicting than other methods. To make the prediction for 1, 3, and 5 years more reliable, a new chart was made. This chart uses both the 4-TRLs model and the patients' basic information. Rooted in the nomogram, a total score of 285 was linked to different chances of living. The rates were 80.8%, 64.9%, and 54.8% at 1, 3, and 5 years, severally (Fig 7C). The calibration curves showed a good match. What the curves predicted was close to what actually happened (Fig 7D). DCA was employed to evaluate the clinical utility of the nomogram. The results revealed that the nomogram provided greater net benefit across a range of threshold probabilities compared to individual clinical and pathological factors, supporting its strong potential for informing clinical decision-making (Figs 7E-G). Moreover, PCA analysis was used to distinguish five gene expression profiles (all genes, TRGs, hub genes, TRLs, and 4-TRLs). The results indicated that when using 4-TRLs to separate patients, those in the H-R cohort were dispersed across multiple quadrants (S4 Fig).

## TMB and TIDE

By comparing the H-R and L-R groups' somatic mutation frequencies, it was observed that the H-R cohort (150/182 [82.42%] samples) exhibited higher mutation frequency in comparison with the L-R cohort (143/179 [79.89%] samples). The top 15 driver mutations are revealed in Figs 8A and B. Chromosomal locations and functional annotations of these mutated genes are detailed in Figs 8C and 8D. Then, rooted in mRNA expression data, the role of 4-TRLs in regulating HCC cancer stem cells (CSCs) was further explored. The results exhibited substantial positive correlation between 4-TRLs and RNA stemness scores in this analysis (RNAss) (Fig 8E, r = 0.25, p = 1.8e-06). TMB can be employed as a biomarker reflecting the number of mutations within tumor cells. The inter-cohort TMBs were evaluated. Patients were divided into two groups. One was the high-TMB (H-TMB) group, and the other was the low-TMB (L-TMB) group. The middle TMB value was used as the standard for this split. But the inter-group difference in TMB was insignificant (Fig 8F, $P > 0.05$). From the Kaplan-Meier analysis, it can be found that the overall survival of patients in the H-TMB group was less favorable than those in the L-TMB group (Fig 8G, P = 0.031). The combined prognostic value of the 4-TRLs signature and TMB was further investigated. Study results showed that patients in the H-R group lived much shorter lives. Their health results were also worse. In fact, much shorter survival time was found in these patients. What's more, those who had both H-R and high TMB status had the shortest total survival time. Their health outcomes were the worst. This suggests that the 4-TRLs mark is better at predicting results. It's more useful than TMB alone (Fig 8H, p < 0.001). Dysfunction and exclusion are combined under the TIDE score system. This index was used to check how likely cancer cells are to avoid the immune system. Compared with the H-R group, the L-R group had much higher TIDE and dysfunction scores. And their exclusion scores were lower (Figs 8I-K, p < 0.001). These results show that the 4-TRLs signature is very good at guessing how HCC patients will respond to immunotherapy.

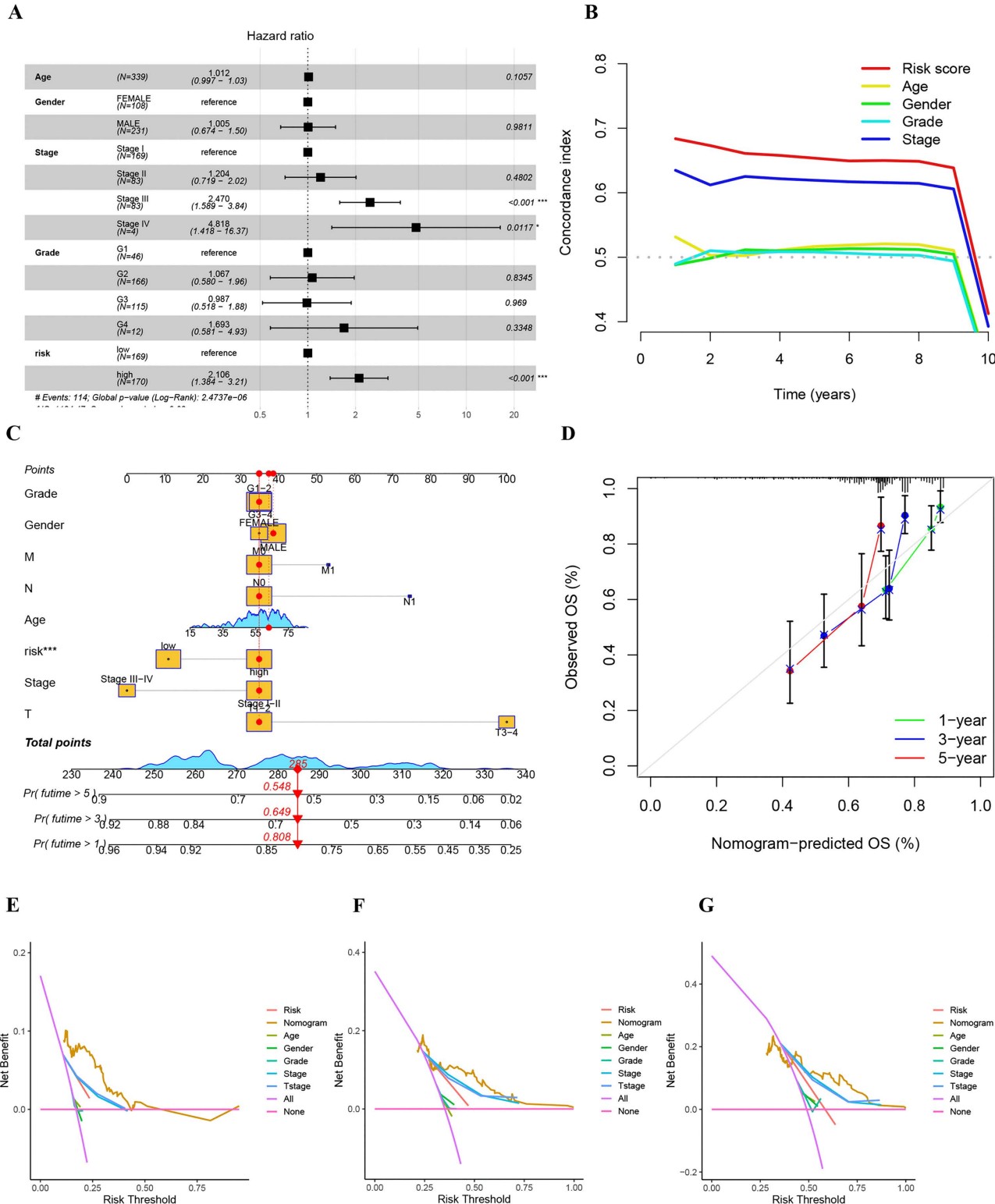

**Fig 7. Nomogram construction and validation.** (A) Forest plot from Cox regression analysis identifying independent prognostic factors. (B) C-index values comparing the predictive performance of the 4-TRLs signature and conventional clinicopathological features. (C) Nomogram integrating the

4-TRLs signature and clinical characteristics for predicting survival probability. (D) Calibration curves evaluating the agreement between predicted and observed survival outcomes. (E-G) Decision curve analysis assessing the clinical utility of the nomogram for predicting overall survival at (E) 1 year, (F) 3 years, and (G) 5 years.

## Tumor immune characteristics and drug sensitivity profiling

The CIBERSORT algorithm was employed to explore the correlation between the 4-TRLs signature and infiltration levels of 22 immune cell types. Plasma cells, T cells CD4 memory activated, T cells follicular helper, T cells regulatory (Tregs), Macrophages M0, Eosinophils, and Neutrophils were more abundant in the H-R cohort, whereas natural killer (NK) cells, B cells naive, T cells CD4 memory resting, Monocytes, Macrophages M2, and Mast cells resting were more prevalent in the L-R cohort (Figs 9A and 9B, S8 Table). These findings were corroborated through parallel analysis with XCELL, TIMER, QUANTISEO, MCPCOUNTER, EPIC, and CIBERSORT-ABS algorithms, which yielded concordant results (S5 Fig, S9 Table). Correlation analysis between the 4-TRLs and immune cell infiltration is presented in Fig 9C. Single-sample gene set enrichment analysis (ssGSEA) revealed enhanced activation of immune-related pathways—comprising antigen-presenting cell (APC) co-stimulation, and major histocompatibility complex (MHC) class I—in the H-R group. But the type II interferon response was found to be weaker (Fig 9D). We compared the gene expression between the two groups. In the H-R group, CTLA4, PDCD-1, CD276, LAG-3 were strikingly up-regulated, as seen in Fig 9E. This suggests that the immune environment in these patients was more active, which might also make it easier to target with treatments. Immunophenotypic stratification revealed a higher prevalence of C3 immune subtypes in the L-R group, whereas the C4 subtype predominated in the H-R cohort (Fig 9F; $p < 0.001$). Drug sensitivity profiling by utilizing the "pRRophetic" R package revealed that H-R patients exhibited increased sensitivity to agents comprising 5-fluorouracil, dasatinib, afatinib, and lapatinib (Figs 9G-J). Conversely, sorafenib, nelarabine, cisplatin, and gemcitabine demonstrated lower half-maximal inhibitory concentration ($IC_{50}$) values in the L-R group (Figs 9K-N). These findings may inform personalized therapeutic strategies for distinct molecular subtypes of hepatocellular carcinoma. All the drugs are displayed in S6 and S7 Figs.

## Cluster analysis

Unsupervised consensus clustering based on CIBERSORT-derived immune cell infiltration profiles classified TCGA-LIHC patients into four distinct clusters: Cluster A (n = 74), Cluster B (n = 108), Cluster C (n = 121), and Cluster D (n = 68) (Fig 10A). Optimal cluster stability and subgroup separation were confirmed by consensus cumulative distribution function (CDF) analysis and relative change in area under the CDF curve, supporting k = 4 as the most appropriate partition (Figs 10B-D). Among the four clusters, the overall survival was more conspicuous when Kaplan–Meier survival analysis was conducted ($p < 0.001$). More favorable survival outcomes can be obtained in Clusters A and B. But poorer prognosis was confirmed in Clusters C and D. Cluster A exhibited the best result, while Cluster D displayed the worst survival (Fig 10E). A heat map showed different ways immune cells spread in each cluster (Fig 10F). We also did a correlation test. It helped make clear how immune cells interact with each other (Fig 10G). As displayed by a comparative evaluation, higher levels of regulatory T cells, follicular helper T cells, activated CD4+memory T cells, and CD8+T cells gathered within Cluster A. Within Cluster B, M2 macrophages, monocytes, and mast cells were highly representative. Within Cluster C, resting CD4+memory T cells conducted a major role. In comparison, Cluster D demonstrated predominant infiltration of M0 macrophages. As immune microenvironment scoring revealed, a significantly more active immune milieu can be detected in Cluster A. This phenomenon was less obvious in other clusters (Fig 10H, $p < 0.001$). Distinct cluster-specific profiles were revealed by differential expression analysis of eight immune checkpoint genes. Cluster A exhibited significant up-regulation of PD-L1, TIGIT, LAG3, IDO1, CTLA4, and BTLA, whereas Cluster D exhibited predominant expression of PD-1 and TIM-3 (Figs 10I-P). The concurrent enrichment of both immune checkpoint molecules (e.g., CD274, LAG3) and

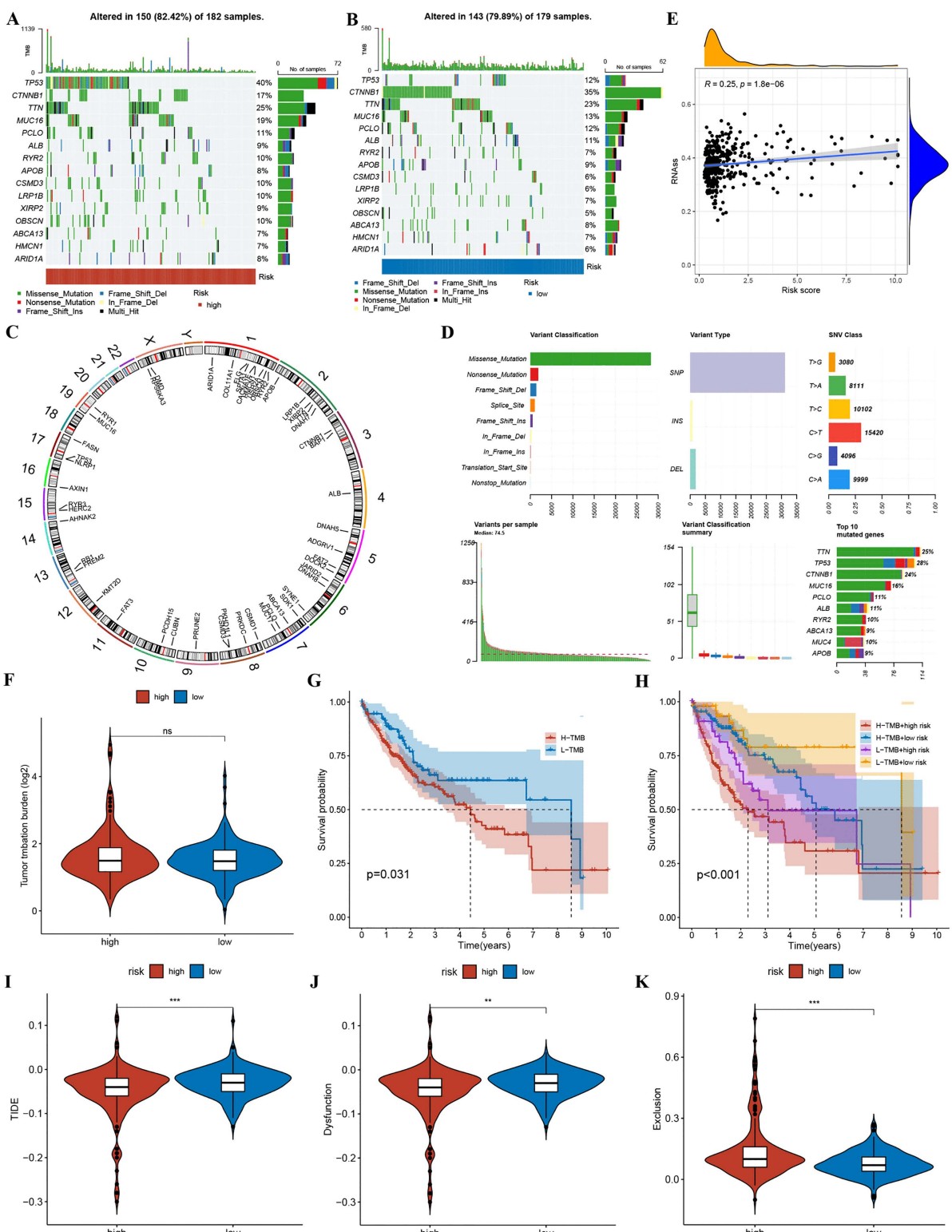

**Fig 8. TMB and TIDE.** (A) Waterfall plot of somatic mutation characteristics in the H-R group. (B) Waterfall plot of somatic mutation characteristics in the L-R group. (C) Chromosomal locations of somatic mutations for the 50 most frequently altered genes in HCC. (D) Comprehensive annotation of mutated

genes in HCC, comprising variant types and functional impacts. (E) Relationship of risk cohorts and CSC index. (F) Violin plot illustrating the differences in TMB between different risk groups. (G) Kaplan-Meier survival curves comparing overall survival between high and low TMB groups. (H) Kaplan-Meier analysis of overall survival across four subgroups stratified by combined TMB status and 4-TRLs risk signature. (I-K) Violin plots displaying differences in TIDE scores (I), dysfunction scores (J), and exclusion scores (K) between H-R and L-R groups.

immunostimulatory infiltrates (e.g., CD8 + T cells, Tregs) in Cluster A provides a potential mechanistic basis for its superior clinical outcomes.

## Single-Cell RNA sequencing data analysis

We integrated single-cell transcriptomic data from five HCC samples in the GSE242889 dataset for subsequent analysis. Subsequent to stringent quality control, retained cells met the following criteria: unique genes detected per cell (nFeature_RNA) ranged between 2,000–6,000, total UMI counts (nCount_RNA) fell within 20,000–80,000, and mitochondrial gene percentage (percent.mito) was below 20% (Fig 11A). Quality assessment revealed a noticeable and positive correlation between sequencing depth and gene detection (Pearson's r = 0.87), and a weak correlation with mitochondrial gene expression (r = 0.21), demonstrating superior data quality and cellular viability (Fig 11B). To find the most active genes, we selected 2,500 from the original set of 21,031 for our next steps (Fig 11C). By adopting uniform manifold approximation and projection (UMAP) dimensionality reduction, we identified 27 distinct cell clusters (Fig 11D). We performed PCA based on these genes. The results showed that the first three principal components were mainly driven by the following genes: VIM, HLA-DRB6, and TYROBP (PC1); AIF1, FCER1G, and TYROBP (PC2); and RGS5, TAGLN, and NDUFA1L2 (PC3) (Fig 11E). Referring to the human primary cell atlas, these clusters were automatically annotated using the SingleR algorithm and classified into eight main cell types: T cells, B cells, liver cells, macrophages, dendritic cells (DCs), tissue stem cells, monocytes, and endothelial cells (Fig 11F). Figs 11G-L illustrate the expression distribution of the 4-TRLs across cellular clusters. Fig 11G's violin plots show their expression patterns across different cell populations. CYTOR showed a specific expression pattern. It was predominantly located in T cells, tissue stem cells, and B cells (Fig 11H). AC026356.1 was predominantly expressed in endothelial cells and macrophages (Fig 11I). AL512598.1 and MIR210HG showed a clear pattern. They were primarily enriched in hepatocytes (Figs 11J and 11K). A gene expression dot plot by cluster also shows 4-TRLs' different expression among main cell types (Fig 11L).

## Pilot experimental validation of the 4-TRLs signature and internal assessment of lncRNA AC026356.1

We performed RT-qPCR on 15 paired HCC and adjacent non-tumor samples to check the 4-TRLs signature first. This analysis compared four lncRNAs. It investigated their expression in both cancerous tissues and healthy ones from the same patients (Figs 12A-D). In analyzing the above results, all four lncRNAs' expression patterns matched our TCGA bioinformatic predictions. Each was significantly up-regulated in tumor tissues (Fig 12E). This finding offers preliminary support for the technical reliability of our computational approach; however, the small cohort size limits any inferential conclusions regarding clinical validity. By focusing on the pivotal lncRNA AC026356.1, the prognostic relevance of the signature was further explored. We used the median expression level as a standard. Then, TCGA-LIHC's HCC samples were divided into H-R and low-expression groups. Kaplan-Meier survival analysis indicated that high AC026356.1 expression was associated with reduced overall survival (Fig 12F, P < 0.001). Analyzes of risk score distribution and survival status suggested that patients in the high-expression group tended to have shorter survival times and higher mortality rates (Figs 12G-H). ROC curve analysis showed that AC026356.1 could discriminate HCC patients from healthy controls (AUC = 0.77, 95% CI 0.72–0.83; Fig 12I). Time-dependent ROC analysis yielded AUC values of 0.71, 0.62, and 0.65 for 1-, 3-, and 5-year overall survival, respectively (Fig 12J). Expression of AC026356.1 was also found to be significantly associated

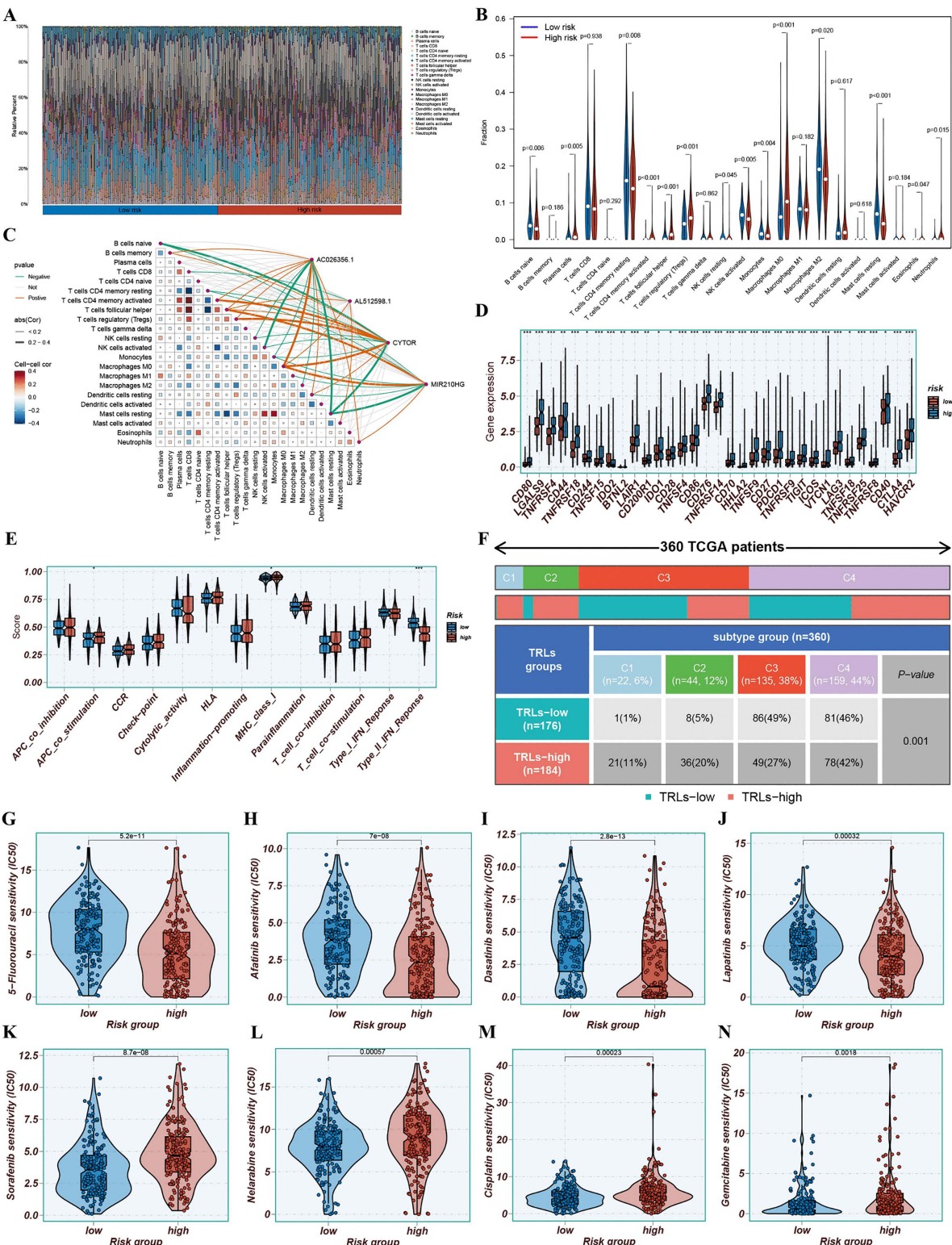

**Fig 9. Tumor immune characteristics and drug sensitivity profiling.** (A, B) Differences in immune cell infiltration between risk groups assessed by adopting the CIBERSORT algorithm, presented as a waterfall plot (A) and violin plots (B). (C) Correlation heatmap between the 4-TRLs signature and 22 immune cell types. (D) Differential immune function activity between risk groups analyzed via single-sample gene set enrichment analysis. (E) Correlation matrix illustrating associations between the 4-TRLs signature and 29 immune checkpoint genes. (F) Distribution of immune subtypes in H-R and L-R groups defined by the 4-TRLs signature. (G-J) Violin plots illustrating differential sensitivity (IC$_{50}$) to 8 clinically relevant anticancer agents between risk groups.

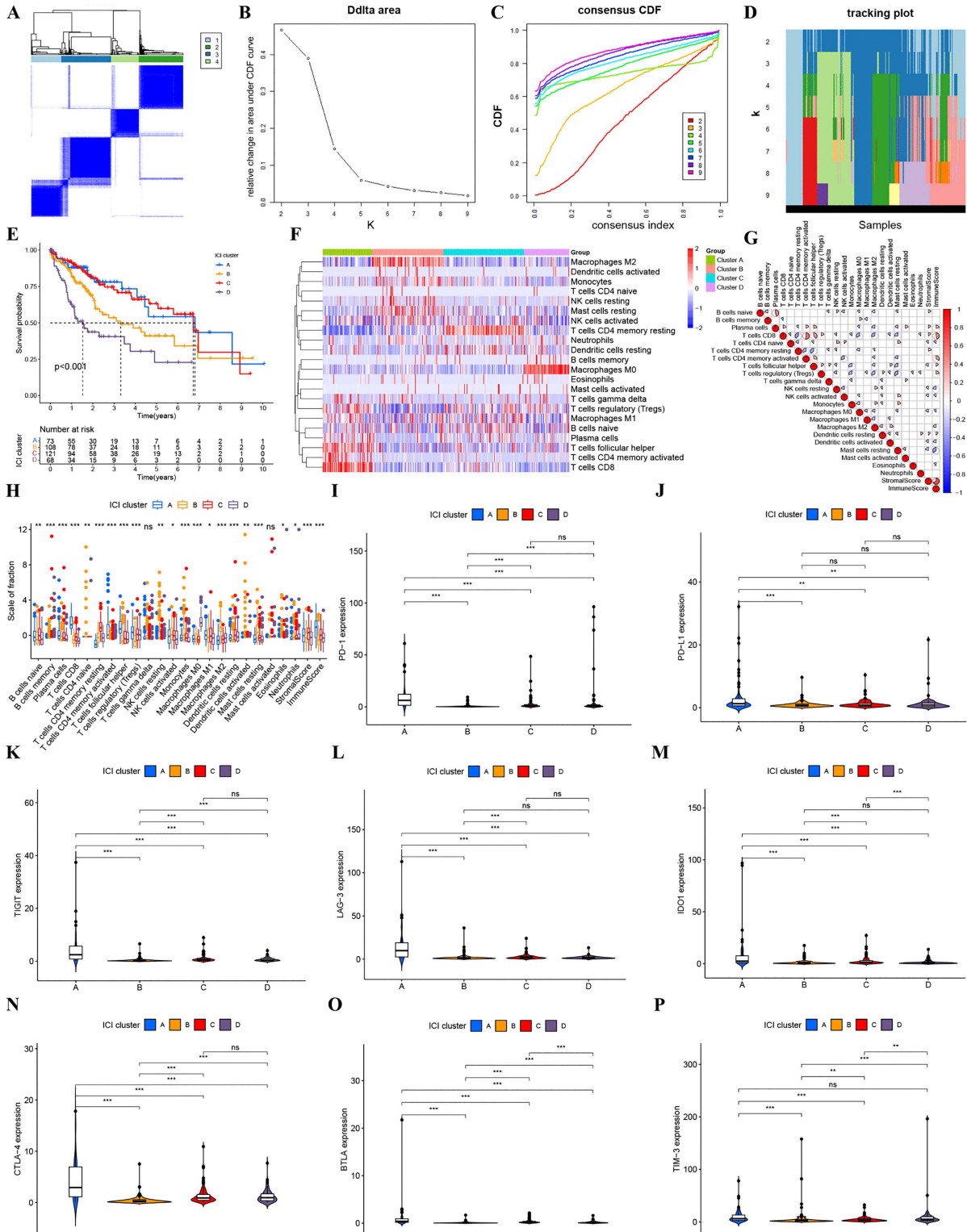

**Fig 10. Cluster analysis.** Consensus clustering matrix at k = 4. (B) Relative changes in the area under CDF curves for k = 2–9. (C) Consensus clustering CDF plot for cluster numbers k = 2–9. (D) Sample clustering tracking graph for k = 2–9. (E) OS of different clusters. (F) Heat map of immune cell infiltration differences among different clusters. (G) Correlation graph among immune cells. (H) Differences in the abundance of 22 immune cells between the four clusters. (I-J) Immune checkpoint genes differences analysis of HCC patients with different clusters.

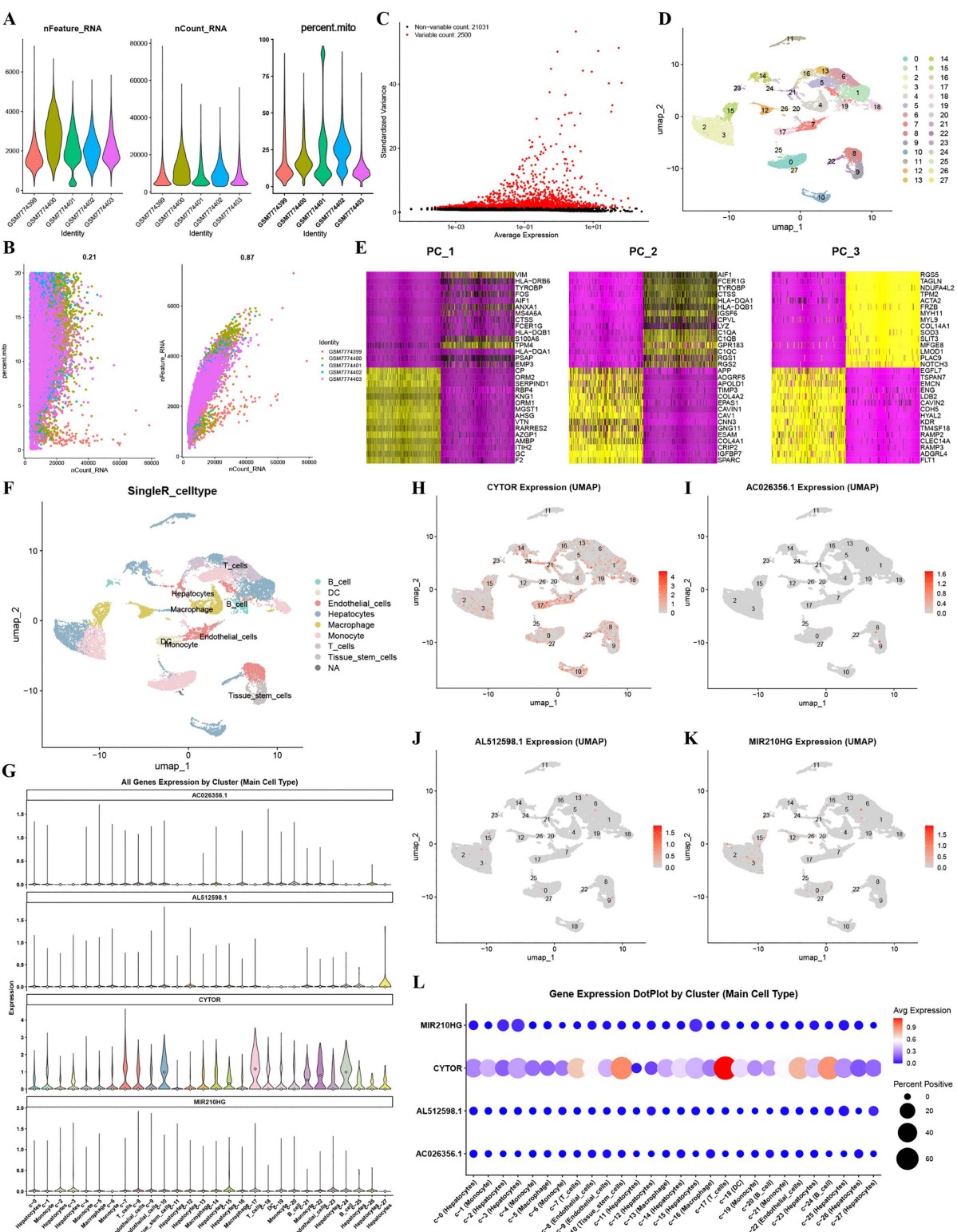

**Fig 11. Single-Cell RNA Sequencing Data Analysis.** (A) Violin plots display gene counts and sequencing depth across cells. (B) Pearson correlation analysis demonstrates negative correlation between sequencing depth and mitochondrial gene expression, and positive association with gene counts.

(C) Volcano plot underlines highly variable genes across the cell population. (D) UMAP nonlinear dimensionality reduction identifies 27 distinct cell clusters. (E) Heatmap suggests the top 30 marker genes for the first three clusters. (F) The 27 cell clusters are annotated into 8 major cell types. (G) Violin plots illustrate expression patterns of the 4-TRLs across different cell populations. (H-K) Distribution of individual TRLs: (H) CYTOR in T cells, tissue stem cells, and B cells, (I) AC026356.1 in endothelial cells and macrophages, (J) AL512598.1 and (K) MIR210HG primarily in hepatocytes. (L) Dot plot summarizes expression levels of the 4-TRLs across major cell types, illustrating both expression intensity and prevalence.

with age, tumor stage, and T category (Fig 12K). Both univariable and multivariable Cox regression analyzes identified AC026356.1 expression and tumor stage as independent prognostic factors for overall survival (Figs 12L-M).

## Prediction of a telomerase related lncRNA-miRNA-mRNA regulatory network

To explore the potential functional mechanisms of the 4-TRLs signature in HCC, we constructed a putative molecular regulatory network. Using StarBase, we predicted a regulatory relationship between AC026356.1 and its target miRNAs (Fig 13A), with the predicted binding sites illustrated in Fig 13B. In the TCGA-LIHC cohort, AC026356.1 expression was significantly up-regulated (Fig 13C), whereas miR-126-5p was down-regulated (Figs 13D–E), and their expression levels showed a significant inverse correlation (Fig 13F). We next identified potential mRNA targets of miR-126-5p through an integrated analysis of multiple databases (StarBase, TargetScan, miRWalk, and mirDIP) (Fig 13G). Four telomerase-related genes—NFIB, SKAP2, PRKAA2, and YAP1—were selected for further investigation (Fig 13H). Fig 13I shows the differential expression patterns of these four genes between HCC and normal samples in the TCGA-LIHC dataset. AC026356.1 expression exhibited significant positive correlations with each of these candidate targets (Figs 13J–M). The expressions of these four TRGs were verified using six independent external validation sets, and the expression differences were consistent with our prediction results (S8 Fig). Taken together, these computational predictions suggest that AC026356.1 could potentially influence hepatocarcinogenesis through a miR-126-5p-mediated regulatory network.

## Discussion

Telomerase functions as a ribonucleoprotein complex that adds repetitive, guanine-rich DNA sequences to the ends of chromosomes. Telomerase contains the catalytic telomerase reverse transcriptase (TERT), the RNA template (TERC), and accessory proteins. These components together contribute to telomere maintenance [23]. The majority of cancer cells achieve cellular immortality. This allows limitless replicative ability. This is in contrast to most somatic cells, which enter senescence and subsequent apoptosis after 50–70 divisions. Cancer cells accomplish this by re-expressing telomerase [24,25]. Thus, telomerase re-expression is frequently a driver of malignant transformation. As we've seen, TERT exerts extensive influence in many cancers. It acts on a variety of molecular pathways. On top of that, it influences the behavior of tumors and patient prognosis. For example, in esophageal adenocarcinoma, the tendency for survival is unfavorable. More specifically, if the TERT gene is subject to amplification, it correlates with poorer survival [26]. In early-stage non-small cell lung cancer, differences are observed in outcome achieved by patients. This is related to genetic variation, particularly in the TERT-CLPTM1L region. Also, high levels of transcript for TERT influence outcome [27]. If TERT is over-expressed, more aggressive behavior can be seen in BRAF-driven thyroid carcinomas, partly through an effect on enhancing the number of ribosomes produced [28]. In bladder cancer, it can be predicted that higher recurrence rates utilizing only TERT promoter mutations. Thus, they represent potential clinical biomarkers [29]. In the context of HCC, TERT expression is elevated, thereby accelerating the growth of tumors. The reason is partly that it tumbles p21 protein degradation. For a further part, mutant TERT promoters can be genome-editable, an idea partly already supported in preclinical studies. The papers above hold that the phenomenon reduces liver cancer [30,31]. Collectively, It can be seen from these observations that the prognostic significance of telomerase in cancer is understudied. Apart from that, we'd like to focus investigation further on telomerase-associated pathways in the future. For one goal is discovery of biomarkers, for another is targeting therapy.

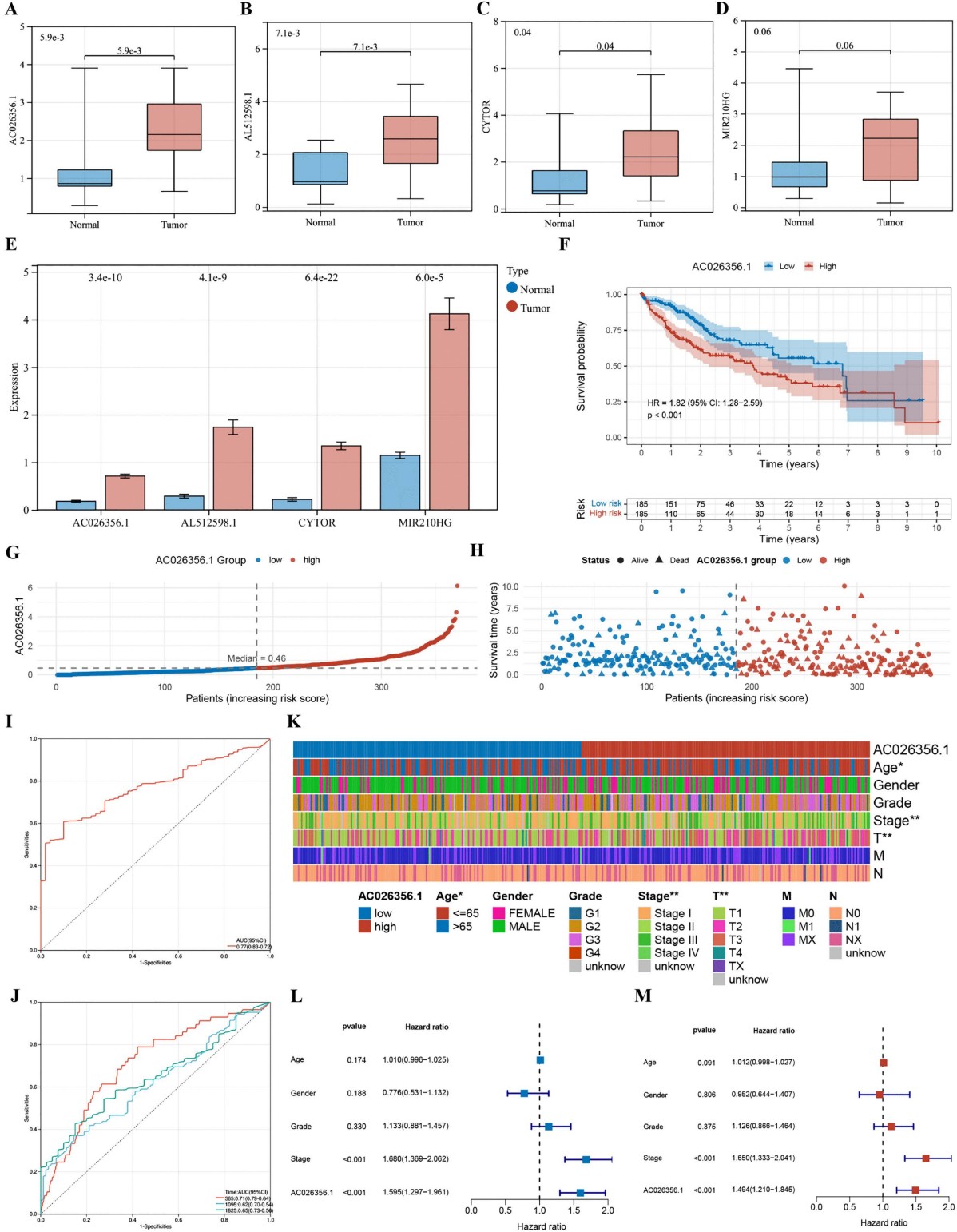

**Fig 12. Pilot experimental validation of the 4-TRLs signature and internal assessment of lncRNA AC026356.1.** (A-D) RT-qPCR analysis of AC026356.1 (A), AL512598.1 (B), CYTOR (C), and MIR210HG (D) expression levels in 15 paired HCC tumor and adjacent normal specimens, compared using Student's t-test. (E) Differential expression of 4-TRLs between tumors and matched benign tissues in TCGA-LIHC cohort. (F) K-M survival

curves stratified by AC026356.1 expression levels. (G) The risk score distribution plot stratified by AC026356.1 expression levels. (H) Survival time and status distribution plot rooted in AC026356.1 expression levels. (I) ROC analysis of AC026356.1's diagnostic accuracy. (J) Time-dependent ROC curves for AC026356.1 high/low expression groups. (K) Heat map illustrating the correlation between AC026356.1 differential expression and clinical features. (L-M) Forest plots of univariate (L) and multivariate (M) Cox regression analyzes.

As proved for dozens of times, lncRNAs can act as the drug and lead to HCC inception and progress. The lncRNA SNHG1, for instance, can promote the growth of HCC in vivo which works through the axis miR7-5p/IGF2 BP2 thus can upregulate IGF2 BP2 [32]. Also, tumor proliferation and migration can be aided by GBAP1 but via METTL3-dependent regulation of the miR-22-3p/BMPR1A/SMAD pathway, GBAP1 can reduce autophagy and apoptosis in Hep3B and MHCC97H cells [33]. On the other hand, LINC01089 can play a role of tumor suppressor and works by interrupting epithelial-mesenchymal transition, which down-regulating DIAPH3 and blocking ERK/Elk1/Snail signaling [34]. Though these examples depict the role of lncRNAs in HCC, still not much is known regarding the participation of TRLs and to fill this gap, we constructed a TRLs-based prognostic signature in predicting survival of HCC patients.

Through our rigorous bioinformatic screening and statistical modeling, we constructed the novel 4-TRLs signature, which was protective and exhibiting superior predictive accuracy and clinical interpretability compared with previous models. Using differential expression analysis of TRGs for the purpose of identifying TRLs, we utilized the TCGA-LIHC datasets and combined them with co-expression analysis of lncRNAs. A 4-TRLs signature exists. It comprises AC026356.1, AL512598.1, CYTOR, and MIR210HG. Univariable and multivariable Cox regression were carried out first. Then feature selection was LASSO regression. Through these, this model was developed accordingly. Supported by ROC curves and C-index analyzes, we found that the model is predictive. It is a major strength of our study that we applied SHAP analysis to the 4-TRLs signature to enhance interpretability and actionable clinical applicability. In our analysis, the top ranked features, as identified by SHAP, such as MIR210HG and CYTOR, are indicative of considerable influence on the prognostic outcome and perhaps clinical risk classification. Features with the greatest feature-level mean positive SHAP values among H-R patients are potential drivers of the unfavorable prognosis. This information may be of prognostic value clinically in a number of ways: first, for patient stratification, clinicians may choose to monitor patients closer, with pre-emotive interventions for those with high levels of H-R features that produce strongly positive SHAP values in our signature; and further, the directionality of SHAP values helps clinicians to understand which stratified H-R patients they should target features for, since certain molecular features may drive their individual outcomes. Furthermore, as revealed in S2 Fig, our model exhibited competitive predictive performance for both 3-year survival (AUC 0.759, versus 0.617–0.789 across other models) and 5-year survival (AUC 0.770, versus 0.630–0.793 across other models) compared to 16 previously published prognostic signatures.

The poor prognosis in the H-R group may also be a consequence of a multifactorial immunosuppressive circuitry where telomerase activity may be the conductor that orchestrates Treg recruitment, NK cell 'normalizing,' and multiple immune checkpoint upregulation. In the tumor microenvironment, telomerase activity has the potential to recruit Tregs to the sites of tumors, and it may facilitate this through the secretion of chemokines by tumor cells which themselves can modulate the expression of multiple cytokines, including CCL22 [35]. In HCC, telomerase "high" tumor cells may also recruit Tregs, through secretion of such chemokines and thereby directly inhibit antitumor immunity through Treg recruitment [36]. Such recruitment may result in the formation of a local immunosuppressive niche that shields tumor cells from effector T-cell responses and provides at least a partial explanation for the poor prognosis of the H-R groups. Moreover, telomerase activity in the HCC tumor microenvironment may inhibit NK cell function, as HCC patients with TERT-mutated tumors were more likely to have reduced proportions of activated NK cells and indicated a relationship between higher tumor telomerase activity and NK cell dysfunction [37]. Increased activity of telomerase within some TERT-mutated tumors reprograms metabolic states in tumor cells (e.g., through upregulation of glycolysis) that may lead to remodeling of the metabolic state of the microenvironment [38]. Such changes in the metabolic microenvironment may induce mitochondrial fission events

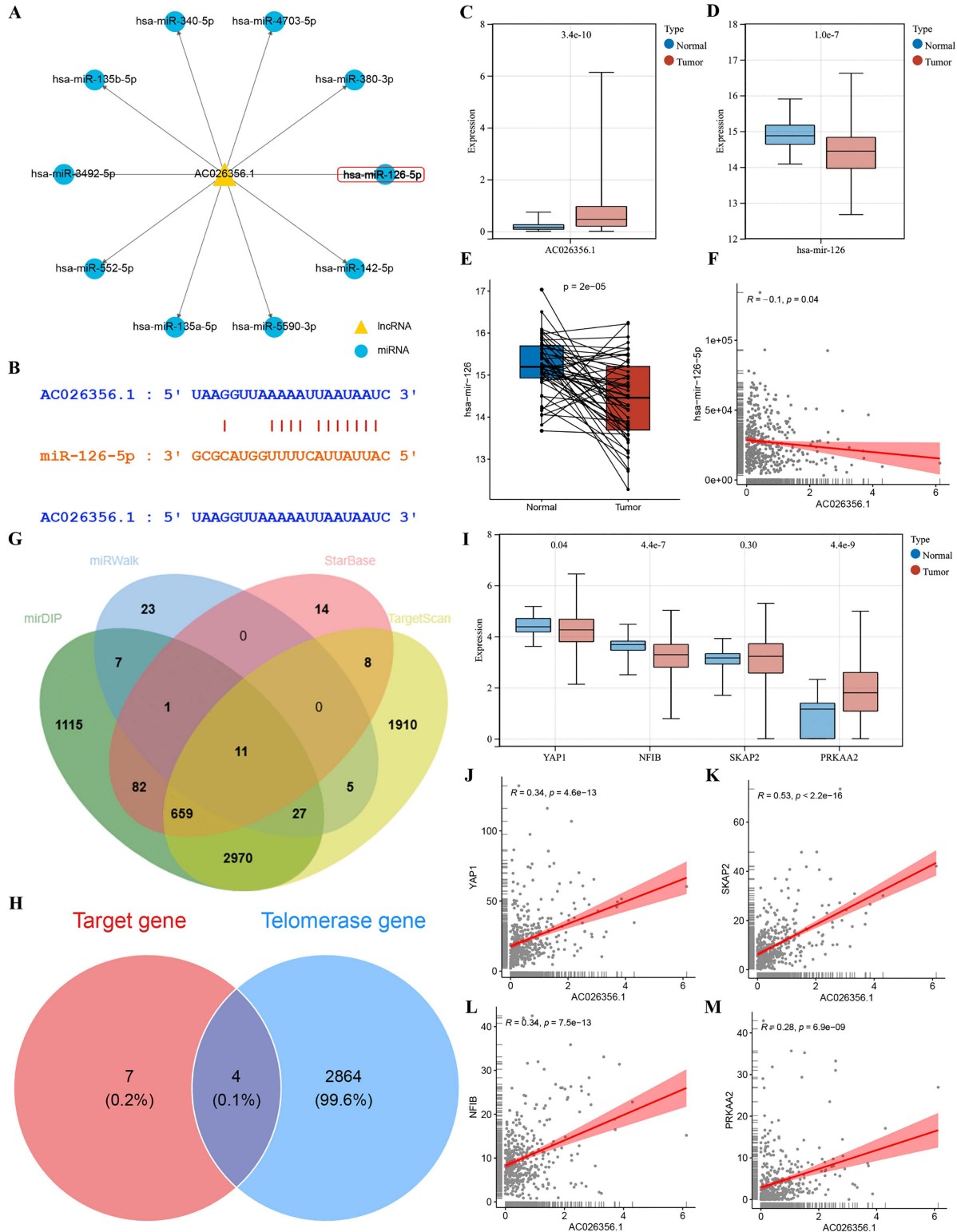

**Fig 13. Prediction of a telomerase-associated lncRNA-miRNA-mRNA regulatory axis.** (A) lncRNA-miRNA interaction network predicted by utilizing StarBase database. (B) Predicted binding sites between the lncRNA AC026356.1 and target miRNA. (C) Box plot illustrating differential expression of AC026356.1 between tumor and normal tissues in the TCGA-LIHC cohort. (D-E) Comparative expression analysis of AC026356.1 in the TCGA-LIHC

dataset: (D) unpaired tumor versus normal tissues; (E) paired tumor and adjacent non-tumorous tissues. (F) Scatter plot illustrating the correlation between AC026356.1 and hsa-mir-126-5p expression. (G) Venn diagram illustrating the overlap of predicted target mRNAs from four databases (mirDIP, miRWalk, StarBase, and TargetScan). (H) Venn diagram identifying four telomerase-associated target mRNAs. (I) Box plots displaying differential expression levels of the four telomerase-related target mRNAs in the TCGA-LIHC dataset. (J-M) Correlation analyzes between AC026356.1 expression and target mRNAs: (J) YAP1, (K) SKAP2, (L) NFIB, and (M) PRKAA2.

in NK cells that deplete their functional cytotoxicity, leading to NK cell dysfunction and subsequent impairment [38]. The 4-TRLs also correlated significantly with immune checkpoints, including CTLA-4, PD-1 (PDCD-1), CD276 (B7-H3), LAG-3, etc. CTLA-4, is utilized in current HCC immunotherapy protocols [39,40]. A pan-cancer analysis indicated that patients with TERT-mutated tumors had a better response to anti-CTLA-4 therapy[37]. We speculate that telomerase activation conferred by TERT mutations may potentially interact with the immune checkpoint-CTLA-4 within these tumors. Further-more, the elevated TIDE scores computed in the H-R group indicates an immune suppressive tumor microenvironment that promotes immune evasion. Two mechanisms underpin this tumor phenotype: impaired T-cell function and their exclusion from the tumor environment-both immune escape mechanisms of resistance to immune checkpoint blockade [41,42]. In liver cancer, this immune escape phenotype is often characterized by an intra-tumoral accumulation of Tregs along with reduced cytotoxic T-cells into the TME. These immunosuppressive factors can reduce the therapeutic efficacy of anti-PD-1/PD-L1 therapy [43,44]. We hereby posit that high telomerase activity may induce a cold tumor microenviron-ment via recruitment of immunosuppressive components and induction of alternative immune checkpoints like LAG-3 and CD276. Finally, drug sensitivity profiling revealed that L-R patients may respond better to sorafenib, nelarabine, cisplatin, and gemcitabine. Sorafenib has been widely used as a first-line treatment for advanced HCC with significant survival ben-efit. In total, these computational and partial experimental findings provide adequate reason for us to further pursue the clinical implications of the 4-TRLs signatures in predicting risk and guiding patient therapy in HCC.

4-TRLs signature are associated with several pro-tumorigenic phenomena, of which proliferation, cell death as well as metabolism are established processes, and are associated functionally with telomerase, represented in example by the predicted AC026356.1/miR-126-5p axis. As our findings show, each of the four TRLs may function as predictive markers in HCC. By distinct pathways, each TRL influences disease processes. For example, lncRNA CYTOR is regarded as a contributor of HCC progression, through multiple molecular pathways. Tumor proliferation may be promoted via CYTOR by targeting the microRNA-125a-5p/LASP1 regulatory axis, as experimental evidence suggests. Cell proliferation and tumor growth may then be enhanced via the miR-125b/SEMA4C pathway. To go it further, CYTOR modulates prolifer-ation, cell cycle, and apoptosis of HCC cells via regulation of the miR-125b-5p/KIAA1522 axis [45–47]. More than just proliferative, CYTOR is associated with overall survival. This works through modulation of autophagy, hypoxia response, cuproptosis, and immunoautophagy processes. Further, CYTOR can promote cancer cell invasion by promoting EMT [48–52]. Compared, lncRNA MIR210HG "participates in various forms of regulated HCC" - angiogenesis, m6A, cupropto-sis, and ferroptosis being typical forms. They have been reported to affect patient survival and immune microenvironment composition, [53–56]. Finally, cuproptosis, disulfidptosis, m$^6$A RNA modification, and cellular senescence is regulated by AC026356.1 [57–60]. This cascades further impacting on prognosis, tumor microenvironment dynamics, and immunotherapy responsiveness. As one contributed Mechanism suggests, HCC progression may be promoted via m$^6$A-modified AC026356.1 via IGF2 BP1–IL11 signaling axis [61]. In TCGA-LIHC, all four TRLs were significantly up-regulated in tumor tissues. Compared to non-tumorous samples. We assessed the general expression pattern in our institutional cohort by RT-qPCR analysis of a pilot set of 15 paired HCC and adjacent normal specimens, and had concor-dant with the above. We then did an internal validation concentrating on AC026356.1, in order to further assess the clinical relevance of the signature. Research has shown that this lncRNA has considerable prognostic discrimination capacity. Significantly, this holds as compared with conventional clinical pathological variables. It has also been established as an independent risk factor. Our findings thus may tentatively validate the overall power of the 4-TRLs model. In pertaining

to HCC pathogenesis, potential lncRNA-miRNA-mRNA regulatory networks were also explored in our research. By using StarBase, a binding site was predicted between AC026356.1 and hsa-miR-126-5p. mirDIP has also been used for target prediction, with TCGA-LIHC showing consistent trends. AC026356.1 was up-regulated, yet mir-126-5p down-regulated in TCGA-LIHC with a ceRNA mechanism. The observation illustrated a significant demonstration of a striking and negative correlation. Target prediction were from StarBase, TargetScan, miRWalk, and mirDIP, with integration. Eleven candidate mRNAs in total were identified which were regulated by miR-126-5p. Among these, four are telomerase associated genes: NFIB, SKAP2, PRKAA2, and YAP1. These were also predicted with high confidence, appearing as "By sponging miR-126-5p and derepressing telomerase-related target genes," our finding thus suggests supports AC026356.1 contributory of hepatocarcinogenesis.

Although the 4-TRLs signature affords good prognostic prediction for HCC survival and treatment response, several caveats must be acknowledged in the study. Firstly, the lack of detailed treatment choice (such as targeted therapies or immune checkpoint inhibitor regimens) in the TCGA cohort has prevented us from assessing whether the model carried prognostic utility within certain treatment subgroups, likely leading to unmeasured confounding which can only be remediated by validation of the model in prospective, multicentre cohorts with better documented treatment history. Secondly, although the four TRLs were found to be differentially expressed by RT-qPCR in a small cohort of 15 paired tissue samples, the overall small size means the robustness and statistical validity of these findings must be taken with caution, and follow up with larger publicly available cohorts is needed. Perhaps most importantly, our results were mainly validated within the TCGA cohort, as such lack of independent external cohorts when validating the clinical applicability of our model, especially a prospective validation set makes finding wide applicability of our model difficult. Fourthly, possible population issues will have to be acknowledged as the TCGA is a Western cohort where most cases are found to have cancers induced by non-viral causes, and such a predominant population may not be representative of our heterogenous global landscape of cancers with viral hepatitis as a cause of HCC. Lastly, while bioinformatic approaches have suggested that these lncRNAs work in crucial death-related pathways like cuproptosis and ferroptosis, their precise roles and molecular mechanisms in HCC remain to be verified. We also acknowledge that while the AC026356.1-miR-126-5p/YAP1 axis appear to hold at least in part a causal role, this still remains in the domain of bioinformatics. Experimental validation via siRNA knockdown, luciferase assays and rescue studies to attune whether this indeed has a functional cascade in HCC is highly crucial. Indeed, this mechanistic aspect is highly important, thus we likely will attempt systematic gain and loss-of-function experiments in follow up studies and possibly do migration and invasion assays in vitro and also in vivo xenograft studies as well, clarifying the underlying biology of the TRLs in HCC progression.

## Conclusion

To sum up, we have identified a prognostic signature b and developed a novel risk model for predicting overall survival in patients with hepatocellular carcinoma. Our initial experimental data from a pilot cohort provides preliminary support for the model.

## Supporting information

**S1 Fig. Functional Enrichment Analyzes of Candidate TRGs.** Enrichment analysis of the five hub genes, with the top 10 significantly enriched terms and pathways from different databases represented in distinct colors. Dot plot illustrates the distribution of hub genes within corresponding biological terms and pathways.
(TIF)

**S2 Fig. A lollipop plot comparing the AUC values of 4-TRLs with those of 16 published HCC prognostic models.**
(A) 1 year AUC. (B) 3 year AUC. (C) 5 year AUC.
(TIF)

**S3 Fig. 4-TRLs for predicting prognosis in HCC patients with different clinical pathological characteristics.** (A-L) Kaplan-Meier survival analysis with log-rank test for patients stratified by (A, B) Age, (C, D) Gender, (E, F) Grade, (G, H) Stage, (I, J) T stage, (K) N stage, and (L) M stage.
(TIF)

**S4 Fig. Principal component analysis.** (A) All genes. (B) TRGs. (C) Hub genes. (D) TRLs. (E) 4-TRLs.
(TIF)

**S5 Fig. Bubble chart of differences in immune cell infiltration.** Seven algorithms (XCELL, TIMER, QUANTISEQ, MCPCOUNTER, EPIC, Cibersort-ABS, and CIBERSORT) were used to compare the infiltration differences of 22 types of immune cells between the high-risk and low-risk groups. A correlation coefficient $> 0$ is regarded as a high degree of infiltration in the high-risk group, while a coefficient $< 0$ is considered as a high degree of infiltration in the low-risk group.
(TIF)

**S6 Fig. Drug sensitivity analysis.** Drugs more sensitive in low-risk group.
(TIF)

**S7 Fig. Drug sensitivity analysis.** Drugs more sensitive in high-risk group.
(TIF)

**S8 Fig. The expression of four target genes was verified by six independent external datasets.** (A) GSE14520 dataset. (B) GSE121248 dataset. (C) GSE76427 dataset. (D) GSE174570 dataset. (E) GSE54236 dataset. (F) GSE17856 dataset.
(TIF)

**S1 Table. 2867 TRGs.**
(XLSX)

**S2 Table. The primer sequence required for the experiment.**
(XLSX)

**S3 Table. Results of Pearson's correlation analysis.**
(XLSX)

**S4 Table. Univariate Cox regression analysis.**
(XLSX)

**S5 Table. Multivariate Cox analysis of five TRLs.**
(XLSX)

**S6 Table. The screening conditions and processes of TRGs/TRLs at key steps.**
(XLSX)

**S7 Table. Locations of AUC values for 16 published models.**
(XLSX)

**S8 Table. The CIBERSORT algorithm was used to score the degree of immune cell infiltration in the TCGA-LIHC cohort samples.**
(XLSX)

**S9 Table. Seven algorithms were used to verify the differences in immune cell infiltration among HCC patients in different risk groups.**
(XLSX)

## Acknowledgments

The authors gratefully acknowledge databases, such as TCGA, for offering convenient access to datasets. The authors utilized ChatGPT solely for the purpose of improving language fluency and readability during the manuscript preparation.

## Author contributions

**Conceptualization:** Yunhao Zhang.

**Data curation:** Runze Yang, Luchao Xing, Chenghao Wang, Songzhuang Xie.

**Funding acquisition:** Jianlei Yuan.

**Project administration:** Jianlei Yuan.

**Software:** Runze Yang, Chenghao Wang.

**Supervision:** Jianlei Yuan.

**Validation:** Luchao Xing.

**Visualization:** Songzhuang Xie.

**Writing – original draft:** Runze Yang.

**Writing – review & editing:** Yunhao Zhang, Jianlei Yuan.

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
