## [Decision Letter · Decision Letter 0]

14 Nov 2025

Dear Dr. Yuan,

Thank you for submitting your manuscript to PLOS ONE. After careful consideration, we feel that it has merit but does not fully meet PLOS ONE’s publication criteria as it currently stands. Therefore, we invite you to submit a revised version of the manuscript that addresses the points raised during the review process.

We look forward to receiving your revised manuscript.

Kind regards,

Zhanzhan Li

Academic Editor

PLOS ONE

Journal Requirements:

3. Please amend the manuscript submission data (via Edit Submission) to include author Jianlie Yuan.

4. Please amend your authorship list in your manuscript file to include author Jianlei Yuan.

5. Please upload a new copy of Figures 1, 2, 3, 8, 10, and 11 as the details are not clear. Please follow the link for more information:  https://journals.plos.org/plosone/s/figures

Additional Editor Comments:

Please note that some comment of Reviewer 2 is  in the attachment.

Reviewers' comments:

Reviewer's Responses to Questions

**Comments to the Author**

1. Is the manuscript technically sound, and do the data support the conclusions?

Reviewer #1: Yes

Reviewer #2: Yes

2. Has the statistical analysis been performed appropriately and rigorously?

Reviewer #1: Yes

Reviewer #2: Yes

3. Have the authors made all data underlying the findings in their manuscript fully available?

Reviewer #1: Yes

Reviewer #2: Yes

4. Is the manuscript presented in an intelligible fashion and written in standard English?

Reviewer #1: Yes

Reviewer #2: Yes

Reviewer #1: This study centers on hepatocellular carcinoma (HCC) and systematically constructs a prognostic model based on four telomerase-related long non-coding RNAs (TRLs). It comprehensively investigates their predictive value in prognosis stratification, immune infiltration, tumor mutation burden (TMB), and drug sensitivity, supported by preliminary RT-qPCR validation. The manuscript is structurally sound, the analytical workflow is clearly presented, and the topic is innovative. The proposed model shows clinical promise; however, revisions are necessary in areas including data source utilization, experimental design, terminology consistency, and mechanistic interpretation.

Major comments

1.The current model is developed and validated entirely based on TCGA internal data, which poses a risk of overfitting. I suggest incorporating independent external datasets, such as those from ICGC or GEO, to enhance the model’s robustness and generalizability.

2.The LASSO regression procedure for feature selection would benefit from greater transparency. The manuscript should clarify whether the optimal lambda value was chosen based on the minimum cross-validation error (min criterion) or the one-standard-error rule (1-SE criterion). This information is critical for evaluating the model's balance between simplicity and predictive accuracy.

3.The SHAP analysis provides intriguing insights into feature importance within the prognostic model. However, the clinical relevance of these findings remains somewhat abstract. A clear explanation should be added regarding how SHAP-derived feature contributions could translate into practical clinical decision-making or patient stratification strategies in hepatocellular carcinoma management.

4.It is suggested to compare the current telomerase-based lncRNA model with other published prognostic models based on m6A-, ferroptosis-, or cuproptosis-related lncRNAs. Such comparisons would help emphasize the novelty and potential advantages of the proposed model.

5.The correlation analysis of clinical characteristics in the TCGA-LIHC cohort reveals "unknown" entries for key prognostic variables—including tumour grade, stage, and TNM classification—as documented in Table 1 and Figures 5B and 12K. Although the distribution of these missing values appears balanced between the training and validation sets, the methods section does not specify whether these cases were included in survival analyses or model construction, nor does it describe the strategy for handling missing data. We recommend that the authors explicitly state their approach to missing data in the "Statistical Analysis" section to enhance methodological transparency and ensure the reproducibility of the reported findings.

6.It is important to note that CIBERSORT relies heavily on the completeness of its reference expression signature and may introduce estimation bias when applied to highly heterogeneous tumor samples. Furthermore, relying exclusively on a single computational approach may not fully capture the complexity of the tumor immune landscape. I recommend that the authors validate these findings using additional well-established algorithms—such as EPIC, MCP-counter, or TIMER—to enhance the robustness and reliability of the immune infiltration estimates. A consensus derived from multiple methodologies would strengthen the credibility of the immunophenotypic associations reported in this study.

7.The observation that high TIDE scores correlate with the high-risk group is noted, but the biological and clinical implications are not sufficiently discussed. The authors should briefly explain how elevated TIDE scores are mechanistically linked to immunotherapy resistance, citing relevant literature to contextualize the findings within the existing immuno-oncology framework.

8.The proposed AC026356.1–miR-126-5p–mRNA regulatory axis presented in Fig 13A-M represents a mechanistically plausible pathway yet remains computationally derived without experimental validation. I recommend that the authors explicitly acknowledge this limitation in the Discussion section, emphasizing the preliminary nature of these bioinformatic predictions and the need for functional validation to establish causal relationships.

9.The manuscript describes "Internal and external validation of the 4-TRLs signature" (Line 601); however, the validation is limited to internal training/test set splits within the TCGA cohort, without inclusion of independent external cohorts—such as those from GEO databases or multi-institutional collaborations—for proper external validation. While RT-qPCR using hospital-derived samples provides valuable experimental confirmation, this constitutes technical validation rather than true external validation in an independent clinical cohort. We recommend revising the terminology to accurately reflect the validation approaches used and conducting comprehensive language polishing throughout to enhance scientific precision and readability.

10.The manuscript exhibits inconsistencies in the reporting of statistical significance, with variations in p-value formatting (e.g., "p < 0.001" versus "P < 0.001") and insufficient documentation of statistical methods in several figures. I recommend standardizing statistical reporting to the format "p < 0.001" throughout the text and explicitly specifying the statistical tests employed (e.g., log-rank test, Student's t-test) in both figure legends and the Methods section.

Minor comment

1.All acronyms should be defined at first use.

2.Figure titles and legends could be refined to be more concise and professionally formatted.

Reviewer #2: This manuscript presents a four-lncRNA telomerase-related signature (AC026356.1, AL512598.1, CYTOR, and MIR210HG) for prognostic stratification in hepatocellular carcinoma (HCC), using TCGA data and limited RT-qPCR validation (n = 15). The concept of telomerase-related lncRNAs (TRLs) is relevant and aligns with current interests in molecular oncology and bioinformatics-driven biomarker research. However, despite its technical rigor, the methodological presentation should be simplified to enhance readability and comprehension. The clinical interpretation needs to be discussed in greater depth, with stronger correlation to real-world clinical application and past references. The manuscript structure also requires revision in accordance with author guidelines to improve clarity and facilitate publication. Furthermore, as the study is retrospective and includes only a small sample number (15 cases), the authors should avoid overestimating their conclusions. The title should reflect this limitation, for example, by indicating it as a pilot or initial study. Minor Revision

**Do you want your identity to be public for this peer review?** For information about this choice, including consent withdrawal, please see our Privacy Policy

Reviewer #1: No

Reviewer #2: No

---

## [Author Response · Author response to Decision Letter 1]

30 Nov 2025

Dear Reviewers:

Thank you for your letter and for the viewers' comments concerning our manuscript entitled "Prognostic stratification in hepatocellular carcinoma using a telomerase-related lncRNA signature derived from TCGA database" (Reference: PONE-D-25-52401). Those comments are all valuable and very helpful for revising and improving our paper, as well as the important guiding significance to our research. We have studied comments carefully and have made corrections which we hope will meet with approval. The main corrections in the paper and the responses to reviewer's comments are as follows:

Reviewer 1

This study centers on hepatocellular carcinoma (HCC) and systematically constructs a prognostic model based on four telomerase-related long non-coding RNAs (TRLs). It comprehensively investigates their predictive value in prognosis stratification, immune infiltration, tumor mutation burden (TMB), and drug sensitivity, supported by preliminary RT-qPCR validation. The manuscript is structurally sound, the analytical workflow is clearly presented, and the topic is innovative. The proposed model shows clinical promise; however, revisions are necessary in areas including data source utilization, experimental design, terminology consistency, and mechanistic interpretation.

Major comments

1.The current model is developed and validated entirely based on TCGA internal data, which poses a risk of overfitting. I suggest incorporating independent external datasets, such as those from ICGC or GEO, to enhance the model’s robustness and generalizability.

Response: We sincerely thank the reviewer for raising this crucial point regarding the general applicability of our predictive model. We fully acknowledge the importance of external validation. However, most publicly available expression matrices lack standardized lncRNA annotation files and often do not include specific lncRNA probes, which prevented us from directly validating the 4-TRLs signature in independent external datasets by obtaining expression matrices for these four TRLs. To proactively address this limitation, we designed an alternative validation strategy using available datasets. Within the proposed lncRNA-miRNA-mRNA regulatory axis, the four TRLs function upstream of key TRGs. We reasoned that validating these downstream TRGs in external cohorts could provide indirect support for the biological relevance of our overall model. Accordingly, we systematically analysed six independent GEO datasets (GSE14520, GSE121248, GSE76427, GSE174570, GSE54236, and GSE17856) to examine the expression patterns of key TRGs within our predicted lncRNA-miRNA-mRNA network (Lines 106-110; 671-673). As shown in S8 Fig, these TRGs demonstrated consistent differential expression across multiple independent cohorts, providing indirect validation for our 4-TRLs signature model. While we acknowledge that this approach does not replace direct validation of the lncRNA signature, we believe that this robust, multi-cohort confirmation of key downstream elements in our model substantially strengthens its generalizability and alleviates concerns regarding overfitting. We thank the reviewer again for their valuable insights.

2.The LASSO regression procedure for feature selection would benefit from greater transparency. The manuscript should clarify whether the optimal lambda value was chosen based on the minimum cross-validation error (min criterion) or the one-standard-error rule (1-SE criterion). This information is critical for evaluating the model's balance between simplicity and predictive accuracy.

Response: We thank the reviewer for their valuable comments regarding the selection of the lambda value in LASSO regression for feature selection. In our study, the optimal lambda value was determined through cross-validation using the cv.glmnet function in R. As shown in our code, the specific implementation was as follows:

cvfit <- cv.glmnet(x, y, family = "cox", maxit = 1000)

coef <- coef(fit, s = cvfit$lambda.min)

The selection was based specifically on the minimum cross-validation error criterion (min criterion), corresponding to cvfit$lambda.min, which identifies the lambda value that minimizes the cross-validation error. Although the one-standard-error rule (1-SE rule, cvfit$lambda.1se) was also visualized in the accompanying figure for completeness, the final model was constructed using lambda.min to ensure optimal predictive accuracy. Therefore, the optimal lambda value in our analysis was selected based on the minimum cross-validation error criterion. This methodological detail has been explicitly stated in the Materials and methods section of our manuscript (Lines 165-167). We appreciate the reviewer's insightful suggestion, which has enabled us to improve the clarity and transparency of our methodology description.

3.The SHAP analysis provides intriguing insights into feature importance within the prognostic model. However, the clinical relevance of these findings remains somewhat abstract. A clear explanation should be added regarding how SHAP-derived feature contributions could translate into practical clinical decision-making or patient stratification strategies in hepatocellular carcinoma management.

Response: We thank the reviewer for this insightful comment regarding the clinical relevance of SHAP analysis in our prognostic model. We have now added the following discussion section (Lines 735-746) to address how SHAP-derived feature contributions could translate into clinical practice for HCC management: "Subsequently, we applied SHAP analysis to the 4-TRLs signature to enhance its interpretability and potential clinical applicability. In our analysis, top-ranking features identified by SHAP, such as MIR210HG and CYTOR, demonstrated substantial influence on prognostic outcomes. Features with consistently positive SHAP values among high-risk patients indicate potential drivers of unfavourable prognosis. This information holds clinical relevance in several aspects: first, for patient stratification, clinicians could prioritise close monitoring and early intervention for patients exhibiting elevated levels of high-risk features with strongly positive SHAP contributions. Second, the directionality of SHAP values helps elucidate which molecular features may be driving individual patient outcomes, potentially informing targeted therapeutic strategies." We sincerely appreciate this valuable suggestion, which has significantly strengthened the clinical translation aspect of our study.

4. It is suggested to compare the current telomerase-based lncRNA model with other published prognostic models based on m6A-, ferroptosis-, or cuproptosis-related lncRNAs. Such comparisons would help emphasize the novelty and potential advantages of the proposed model.

Response: We sincerely appreciate the reviewer's valuable suggestion regarding the comparative analysis of our 4-TRLs model with other prognostic signatures. As shown in S2 Fig, we compared our 4-TRLs signature with 16 published models (including m6A-, ferroptosis-, and cuproptosis-related signatures) in the TCGA cohort. The results revealed that our 4-TRLs signature achieved superior predictive performance for 3-year (AUC = 0.759) and 5-year (AUC = 0.770) survival, outperforming all other signatures, while maintaining competitive accuracy for 1-year prediction (AUC = 0.744) (Lines 385-392). Furthermore, detailed information for each of the 16 published models—including PMID, figure panel location, biological function, model abbreviation, and AUC values at 1-, 3-, and 5-years—is systematically compiled in S7 Table. We thank the reviewer for this insightful suggestion, which has strengthened the novelty and translational significance of our study.

5. The correlation analysis of clinical characteristics in the TCGA-LIHC cohort reveals "unknown" entries for key prognostic variables—including tumour grade, stage, and TNM classification—as documented in Table 1 and Figures 5B and 12K. Although the distribution of these missing values appears balanced between the training and validation sets, the methods section does not specify whether these cases were included in survival analyses or model construction, nor does it describe the strategy for handling missing data. We recommend that the authors explicitly state their approach to missing data in the "Statistical Analysis" section to enhance methodological transparency and ensure the reproducibility of the reported findings.

Response: We sincerely thank the reviewer for raising this important methodological point regarding the handling of missing clinical data. We appreciate the opportunity to clarify our analytical approach, and we have now updated the "Statistical analysis" section accordingly to enhance methodological transparency (Lines 268-273). In our study, we adopted distinct strategies for handling missing clinical data in different analytical contexts, based on specific analytical objectives: For correlation analyses (presented in Table 1, Figure 5B, and Figure 12K), samples with "unknown" entries for tumour grade, stage, and TNM classification were retained to provide a comprehensive overview of the entire cohort's clinical characteristics. For prognostic model construction, we included all available HCC samples (N=370) after excluding four cases with zero survival time. Samples with "unknown" clinical features were retained in this analysis, as our LASSO-Cox model utilizes only gene expression data for risk prediction and does not require complete clinical annotation. For survival analysis in S3 Fig, we excluded samples with "unknown" clinical features when assessing the model's stratification capacity across specific clinical subgroups. This approach ensures clear interpretation of Kaplan-Meier curves by maintaining well-defined clinical categories. This stratified approach to missing data allows us to both maximize statistical power in model development and maintain analytical rigor in clinical correlation assessments. We have now explicitly detailed these procedures in the revised manuscript to ensure full reproducibility of our findings. We are grateful for this constructive suggestion, which has significantly improved the clarity and rigor of our methodological description.

6. It is important to note that CIBERSORT relies heavily on the completeness of its reference expression signature and may introduce estimation bias when applied to highly heterogeneous tumor samples. Furthermore, relying exclusively on a single computational approach may not fully capture the complexity of the tumor immune landscape. I recommend that the authors validate these findings using additional well-established algorithms—such as EPIC, MCP-counter, or TIMER—to enhance the robustness and reliability of the immune infiltration estimates. A consensus derived from multiple methodologies would strengthen the credibility of the immunophenotypic associations reported in this study.

Response: We sincerely thank the reviewer for raising this important methodological consideration regarding immune infiltration estimation. We fully agree that relying on a single computational approach may not adequately capture the complexity of the tumor immune microenvironment. In response to this valuable suggestion, we have comprehensively analyzed immune cell infiltration differences between risk groups using seven distinct, well-established algorithms — including not only CIBERSORT but also XCELL, TIMER, QUANTISEO, MCPCOUNTER, EPIC, and CIBERSORT-ABS (Lines 202-204; 523-526). As shown in the bubble plot presented in S5 Fig, the consistent patterns observed across multiple algorithms validate the reliability of our immune infiltration estimates. We have now explicitly described this multi-algorithm validation strategy in the revised Result section and included the comparative results in the S9 Table. We are grateful for this insightful suggestion, which has substantially improved the robustness and reliability of our findings.

7. The observation that high TIDE scores correlate with the high-risk group is noted, but the biological and clinical implications are not sufficiently discussed. The authors should briefly explain how elevated TIDE scores are mechanistically linked to immunotherapy resistance, citing relevant literature to contextualize the findings within the existing immuno-oncology framework.

Response: We thank the reviewer for this insightful comment. In response, we have added a dedicated paragraph in the Discussion section to elaborate on the mechanistic and clinical implications of the elevated TIDE scores in our H-R group. Specifically, we have cited recent literature to contextualize how high TIDE scores reflect key mechanisms of immunotherapy resistance, such as T-cell dysfunction and exclusion, and linked these findings to the established immuno-oncology framework in HCC (Lines 775-782). We appreciate the reviewer's valuable suggestion, which has helped strengthen our manuscript.

8. The proposed AC026356.1–miR-126-5p–mRNA regulatory axis presented in Fig 13A-M represents a mechanistically plausible pathway yet remains computationally derived without experimental validation. I recommend that the authors explicitly acknowledge this limitation in the Discussion section, emphasizing the preliminary nature of these bioinformatic predictions and the need for functional validation to establish causal relationships.

Response: We thank the reviewer for raising this important point regarding the need to acknowledge the computational nature of our proposed regulatory axis. In response to this comment, we have explicitly acknowledged this limitation in the Discussion section of our revised manuscript. Specifically, we have added the following statement: Lastly, while bioinformatic approaches have suggested that these lncRNAs work in crucial death-related pathways like cuproptosis and ferroptosis, their precise roles and molecular mechanisms in HCC remain to be verified. We also acknowledge that while the AC026356.1-miR-126-5p/YAP1 axis appear to hold at least in part a causal role, this still remains in the domain of bioinformatics. Experimental validation via siRNA knockdown, luciferase assays and rescue studies to attune whether this indeed has a functional cascade in HCC is highly crucial (Lines 848-854). We are grateful to the reviewer for this valuable suggestion, which has helped improve the transparency and rigor of our study.

9. The manuscript describes "Internal and external validation of the 4-TRLs signature" (Line 601); however, the validation is limited to internal training/test set splits within the TCGA cohort, without inclusion of independent external cohorts—such as those from GEO databases or multi-institutional collaborations—for proper external validation. While RT-qPCR using hospital-derived samples provides valuable experimental confirmation, this constitutes technical validation rather than true external validation in an independent clinical cohort. We recommend revising the terminology to accurately reflect the validation approaches used and conducting comprehensive language polishing throughout to enhance scientific precision and readability.

Response: We sincerely thank the reviewer for their insightful comments regarding the validation terminology used in our manuscript. We acknowledge that while the RTqPCR validation using hospital-derived samples provides valuable technical confirmation, it does not constitute true external validation in an independent clinical cohort. Accordingly, we have reframed this section to emphasize its role in pilot experimental verification rather than external validation. The subsection heading has been revised to: "Pilot experimental validation of the 4-TRL signature and internal assessment of lncRNA AC026356.1 (Lines 622-623)." In addition, we have performed comprehensive language polishing throughout the manuscript to improve scientific precision and readability. We are grateful for this constructive suggestion, which has significantly improved the accuracy and clarity of our methodological descriptions.

10.The manuscript exhibits inconsistencies in the reporting of statistical significance, with variations in p-value formatting (e.g., "p < 0.001" versus "P < 0.001") and insufficient d

---

## [Decision Letter · Decision Letter 1]

7 Dec 2025

Prognostic stratification in hepatocellular carcinoma using a telomerase-related lncRNA signature derived from TCGA database

PONE-D-25-52401R1

Dear Dr. Yuan,

We’re pleased to inform you that your manuscript has been judged scientifically suitable for publication and will be formally accepted for publication once it meets all outstanding technical requirements.

Kind regards,

Zhanzhan Li

Academic Editor

PLOS One

Additional Editor Comments (optional):

Reviewers' comments:

Reviewer's Responses to Questions

**Comments to the Author**

Reviewer #1: All comments have been addressed

Reviewer #2: All comments have been addressed

2. Is the manuscript technically sound, and do the data support the conclusions?

Reviewer #1: Yes

Reviewer #2: Yes

3. Has the statistical analysis been performed appropriately and rigorously?

Reviewer #1: Yes

Reviewer #2: Yes

4. Have the authors made all data underlying the findings in their manuscript fully available?

Reviewer #1: Yes

Reviewer #2: Yes

5. Is the manuscript presented in an intelligible fashion and written in standard English?

Reviewer #1: Yes

Reviewer #2: Yes

Reviewer #1: All comments have been addressed very carefully. I have no other comment. This manuscript may be acceptable for publication.

Reviewer #2: (No Response)

**Do you want your identity to be public for this peer review?** For information about this choice, including consent withdrawal, please see our Privacy Policy

Reviewer #1: No

Reviewer #2: No

---

## [Editor Report · Acceptance letter]

PONE-D-25-52401R1

PLOS One

Dear Dr. Yuan,

I'm pleased to inform you that your manuscript has been deemed suitable for publication in PLOS One. Congratulations! Your manuscript is now being handed over to our production team.

Kind regards,

on behalf of

Dr. Zhanzhan Li

Academic Editor

PLOS One